# Novel mechanism of metabolic co-regulation coordinates the biosynthesis of secondary metabolites in *Pseudomonas protegens*

Qing Yan[1], Benjamin Philmus[2], Jeff H Chang[1], Joyce E Loper[1,3]*

[1]Department of Botany and Plant Pathology, Oregon State University, Corvallis, United States; [2]Department of Pharmaceutical Sciences, Oregon State University, Corvallis, United States; [3]US Department of Agriculture, Agricultural Research Service, Horticultural Crops Research Laboratory, Corvallis, United States

**Abstract** Metabolic co-regulation between biosynthetic pathways for secondary metabolites is common in microbes and can play an important role in microbial interactions. Here, we describe a novel mechanism of metabolic co-regulation in which an intermediate in one pathway is converted into signals that activate a second pathway. Our study focused on the co-regulation of 2,4-diacetylphloroglucinol (DAPG) and pyoluteorin, two antimicrobial metabolites produced by the soil bacterium *Pseudomonas protegens*. We show that an intermediate in DAPG biosynthesis, phloroglucinol, is transformed by a halogenase encoded in the pyoluteorin gene cluster into mono- and di-chlorinated phloroglucinols. The chlorinated phloroglucinols function as intra- and inter-cellular signals that induce the expression of pyoluteorin biosynthetic genes, pyoluteorin production, and pyoluteorin-mediated inhibition of the plant-pathogenic bacterium *Erwinia amylovora*. This metabolic co-regulation provides a strategy for *P. protegens* to optimize the deployment of secondary metabolites with distinct roles in cooperative and competitive microbial interactions.

*For correspondence: joyce.loper@oregonstate.edu

**Competing interests:** The authors declare that no competing interests exist.

## Introduction

Many microorganisms produce multiple secondary metabolites with diverse chemical properties and ecological functions that contribute to the fitness of the producing organism in natural environments (*Haas and Keel, 2003*; *O'Brien and Wright 2011*, *Traxler and Kolter, 2015*). The biosynthesis of different secondary metabolites by a microorganism is often coordinated (*Bosello et al., 2011*; *Tsunematsu et al., 2013*; *Hidalgo et al., 2014*; *Vingadassalon et al., 2015*; *Cano-Prieto et al., 2015*; *Cui et al., 2016*). Co-regulation of different secondary metabolites is thought to be selected during microbial evolution because it confers competitive advantages to the producing organism in microbial interactions (*Challis and Hopwood, 2003*).

Among the secondary metabolites, antibiotics are particularly intriguing because of their roles in both cooperative and competitive interactions among microorganisms in natural habitats (*Yim et al., 2007*; *Raaijmakers and Mazzola, 2012*). This study focuses on 2,4-diacetylphloroglucinol (DAPG) and pyoluteorin, two broad-spectrum antibiotics with toxicity against fungi, bacteria, oomycetes and plants (*Ohmori et al., 1978*; *Keel et al., 1992*). Certain strains of *Pseudomonas* spp. produce DAPG but not pyoluteorin (*Loper et al., 2012*) and some strains of *Pseudomonas aeruginosa* produce pyoluteorin but not DAPG (*Ohmori et al., 1978*). To date, only a subset of strains of the species *Pseudomonas protegens* are known to produce both compounds (*Loper et al., 2012*). Among these

**eLife digest** Bacteria live almost everywhere on Earth and often compete with one another for limited resources, like space or nutrients. Certain bacteria produce molecules that are toxic to other microorganisms to give themselves a competitive advantage. These toxic molecules are more commonly referred as antibiotics, and are perhaps best known for their importance in medicine. Yet, antibiotics benefit the bacteria that produce them in other ways too. Some bacteria, for example, use antibiotics as chemical signals to communicate with one another and coordinate their activities.

Some bacteria produce many antibiotics with different toxic and signaling activities. These bacteria often coordinate the production of different antibiotics such that the production of one antibiotic shuts down the production of another. This kind of coordination would allow the bacterium to focus its energy on producing only the antibiotic that gives it a competitive advantage at that time. Yet, in most cases, it was not known how the bacterial cell coordinates the production of two different antibiotics.

*Pseudomonas protegens* is a species of bacteria that lives in soil, and produces many antibiotics that are toxic to other bacteria or fungi. The antibiotics are made via distinct pathways of chemical reactions that are catalyzed by different enzymes. However, the production of two antibiotics, called 2,4-diacetylphloroglucinol and pyoluteorin, is tightly coordinated in some strains of *P. protegens*. Now, Yan et al. have discovered how *P. protegens* coordinates the production of these two antibiotics. It turns out that the bacterium produces an enzyme that adds chlorine atoms onto one of the intermediate building blocks used to make 2,4-diacetylphloroglucinol. These "chlorinated derivatives" then activate the genes required to make the second antibiotic, pyoluteorin. The derivatives also signal to other *P. protegens* cells and trigger them to produce pyoluteorin too. Lastly, Yan et al. confirmed that pyoluteorin could inhibit the growth of another species of bacteria called *Erwinia amylovora*.

These new findings highlight an important role played by chemicals that might have previously been considered as merely stepping stones in other biochemical reactions. An important challenge for the future will be to evaluate if other microbes use chemical intermediates in similar ways. Understanding the natural role of more antibiotics and their intermediates should help us to more wisely use existing antibiotics, and might eventually lead to new treatments for infections in humans and other animals.

DAPG- and pyoluteorin-producing strains is the soil bacterium *P. protegens* Pf-5. In the genome of Pf-5, genes for DAPG and pyoluteorin biosynthesis are present in two different gene clusters (*Figure 1A and C*) separated by 3.7-megabases (*Paulsen et al., 2005*). The biosynthetic substrates and intermediates of the two pathways are distinct (*Figure 1B and D*). DAPG biosynthesis begins with synthesis of phloroglucinol (PG) from three molecules of malonyl-CoA by the type III polyketide synthase PhlD (*Achkar et al., 2005*). PG is then acetylated through the action of PhlABC to form DAPG (*Bangera and Thomashow, 1999*). Pyoluteorin biosynthesis begins with activation of L-proline by the L-prolyl-AMP ligase PltF, and attachment to the peptidyl-carrier protein PltL (*Thomas et al., 2002*). The Pro-*S*-PltL intermediate is oxidized by PltE and chlorinated by PltA to yield 4,5-dichloropyrrolyl-*S*-PltL (*Thomas et al., 2002*; *Dorrestein et al., 2005*). The dichloropyrrole is extended by the type I polyketide synthase PltBC followed by release and formation of the resorcinol, presumably catalyzed by PltG to yield pyoluteorin (*Nowak-Thompson et al., 1997*, *1999*).

Despite the independent biochemical and genetic determinants for their biosynthesis, production of DAPG and pyoluteorin by *P. protegens* is tightly coordinated. Mutations in pyoluteorin biosynthetic genes result in loss of pyoluteorin production along with overproduction of DAPG (*Brodhagen et al., 2004*; *Quecine et al., 2016*), and addition of pyoluteorin to bacterial cultures represses the expression of certain DAPG biosynthetic genes and the production of DAPG (*Schnider-Keel et al., 2000*; *Baehler et al., 2005*). Conversely, PG, an intermediate in DAPG biosynthesis, has a concentration-dependent influence on expression of pyoluteorin biosynthetic genes and production of pyoluteorin: nanomolar concentrations of PG are required for pyoluteorin

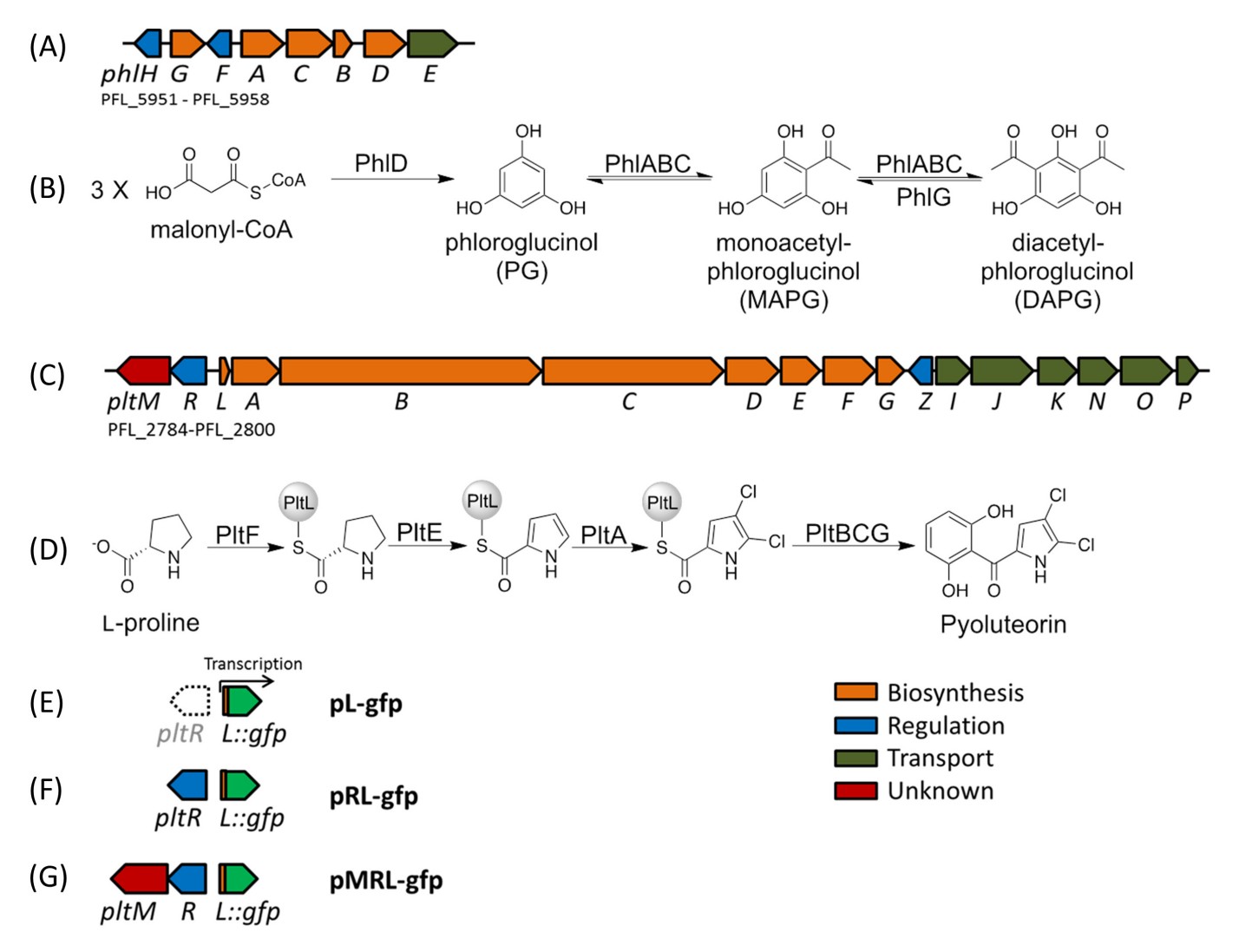

**Figure 1.** Biosynthetic gene clusters for 2,4-diacetylphloroglucinol (DAPG) (A), and pyoluteorin (C), the proposed biosynthetic pathways of DAPG (B) and pyoluteorin (D) of *P. protegens* Pf-5, and the GFP reporter constructs (E, F, G) used in this work. Arrows represent gene location and orientation in the biosynthetic gene clusters, and are colored according to their functions. The open arrow in (E) indicates that *pltR* is not included in the pL-gfp construct.

production but micromolar concentrations of PG repress pyoluteorin production (*Kidarsa et al., 2011*; *Clifford et al., 2016*).

The essential role of PG in pyoluteorin production became evident when we discovered that a *phlD* mutant, which lacks the type III polyketide synthase PhlD responsible for PG biosynthesis, produces neither pyoluteorin nor DAPG. The *phlD* mutant can be complemented for pyoluteorin production by addition of nanomolar concentrations of PG to the culture medium (*Kidarsa et al., 2011*). Furthermore, addition of nanomolar concentrations of PG induces expression of pyoluteorin biosynthetic genes (*Clifford et al., 2016*), indicating that the influence of PG on pyoluteorin production is mediated through gene regulation. Pyoluteorin itself also induces expression of pyoluteorin biosynthetic genes (*Brodhagen et al., 2004*; *Li et al., 2012*). The pyoluteorin biosynthetic genes are expressed under the positive control of PltR (*Brodhagen et al., 2004*) (*Figure 1*), a transcriptional regulator belonging to the LysR family. Both PG and pyoluteorin are known to induce PltR-mediated transcription of pyoluteorin biosynthetic genes, but the relative importance of the two small molecules in inducing gene expression has not been evaluated.

The metabolic co-regulation between DAPG and PLT biosynthetic pathways differs from known examples of metabolic co-regulation between secondary metabolic pathways, which can be classified into four types based on the underlying mechanisms. The first type of co-regulation involves one or more intermediates that are shared by different biosynthetic pathways (*Vingadassalon et al., 2015*; *Cano-Prieto et al., 2015*). The second type results when a single enzyme is shared by different pathways (*Lazos et al., 2010*; *Tsunematsu et al., 2013*). The third type of co-regulation is mediated through functionally redundant enzymes present in different pathways (*Lautru et al., 2007*). The fourth type occurs when the biosynthetic genes for two pathways are controlled by the same 'pathway-specific' regulator (*Pérez-Llarena et al., 1997*; *Bergmann et al., 2010*). Overall these mechanisms of metabolic co-regulation involve a shared biosynthetic intermediate, enzymatic activity, or regulator, but none of these features is shared by the DAPG and pyoluteorin pathways (*Figure 1*).

Here, we describe a mechanism of metabolic co-regulation that is distinct from all described previously. Our results show that PG, an intermediate in DAPG biosynthesis, is transformed by a halogenase encoded in the pyoluteorin gene cluster into chlorinated derivatives that function as cell-cell communication signals inducing expression of pyoluteorin biosynthetic genes. Our work reveals a new mechanism of co-regulation between two secondary metabolic pathways, which is mediated by an intermediate in one pathway that is converted into signals that activate the second pathway.

## Results

### PG is required for PltR-mediated activation of *pltL* expression

To determine the differences between the regulatory roles of PG and pyoluteorin in the production of pyoluteorin, we evaluated the effects of both metabolites on expression of the pyoluteorin biosynthetic gene *pltL* using pL-gfp (*Figure 1E*), a reporter construct containing a transcriptional fusion of the promoter of *pltL* to a promoterless *gfp* (*Yan et al., 2016*). GFP expressed from pL-gfp was monitored in a Δ*pltA*Δ*phlD* double mutant grown in a medium amended with each of the two metabolites alone or in combination (*Figure 2*). In the non-amended medium, and relative to wild-type Pf-5, GFP expression was significantly lower in the Δ*pltA*Δ*phlD* mutant ($p$=2.76E-4), indicating only background expression of *pltL*. This result was expected because of the deletion of *pltA* and *phlD*, which encode enzymes for biosynthesis of pyoluteorin and PG, respectively (*Figure 1*); both pyoluteorin and PG are known to induce expression of pyoluteorin biosynthetic genes (*Brodhagen et al., 2004*; *Clifford et al., 2016*). In the medium amended with 10 nM PG, GFP expression of the Δ*pltA*Δ*phlD* mutant increased nearly tenfold relative to expression levels in non-amended medium, indicating induction of *pltL*. In contrast, addition of 100 nM pyoluteorin failed to alter GFP expression by the Δ*pltA*Δ*phlD* mutant. However, when both pyoluteorin and PG were added, the GFP expression was significantly higher than under treatment with PG alone ($p$=1.44E-4). Therefore, pyoluteorin was not required for *pltL* expression but further induced *pltL* expression in the presence of PG. As expected based on the necessity of PltR for expression of the pyoluteorin biosynthetic genes (*Brodhagen et al., 2004*; *Li et al., 2012*), *pltR* was required for expression of *pltL* and the presence of PG could not restore *pltL* expression in a Δ*pltA*Δ*phlD*Δ*pltR* mutant (*Figure 2*). These results indicated that PG, not pyoluteorin, is required for PltR-mediated activation of *pltL*.

### PG activates expression of *pltL* indirectly

To test whether PG directly or indirectly regulates expression of the pyoluteorin biosynthetic genes, we assessed the effect of PG on expression of *pltL* in *P. fluorescens* SBW25. Like *P. protegens* Pf-5, strain SBW25 is a member of the *P. fluorescens* group but, unlike Pf-5, SBW25 has neither the *phl* gene cluster nor the *plt* gene cluster (*Silby et al., 2009*). Because SBW25 does not have *pltR*, which is required for *pltL* expression, we made the transcriptional reporter construct pRL-gfp, which contains *pltR* and the intergenic region between *pltR* and *pltL* that includes the promoter of *pltL* fused to a promoterless *gfp* (*Figure 1F*). Although PG was sufficient to induce *pltL* expression in Pf-5 (*Figure 2*), PG failed to induce GFP expression from pRL-gfp in *P. fluorescens* SBW25 (*Figure 3A*) and is therefore not likely to function as a signal directly. One possible explanation for these results is that Pf-5 processes PG to another compound, which then directly induces pyoluteorin gene expression.

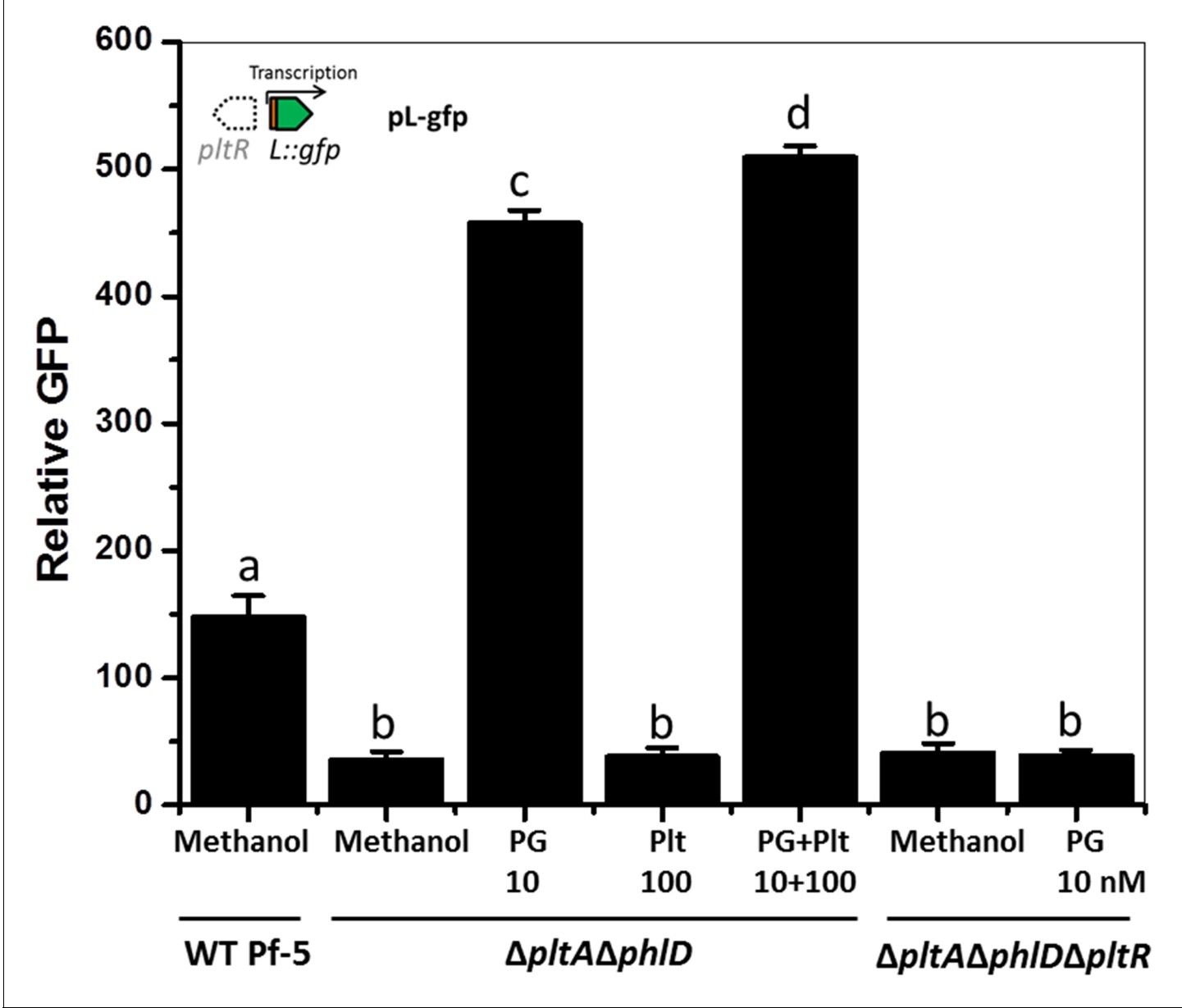

**Figure 2.** Effect of phloroglucinol (PG) and pyoluteorin on expression of *pltL::gfp* in *P. protegens* Pf-5. Pf-5 wild-type and derivatives containing pL-gfp were treated with PG (10 nM) and/or pyoluteorin (100 nM), as indicated. Expression levels of *pltL::gfp* were measured and recorded as relative GFP (fluorescence of GFP divided by $OD_{600}$). Letters above columns indicate treatments significantly different from one another, as determined by ANOVA analysis ($p < 0.05$). Data are means of at least three biological replicates from a representative experiment repeated three times with similar results, and error bars represent the standard deviation of the mean.

The following source data is available for figure 2:

**Source data 1.** Expression of *pltL::gfp* by Pf-5 wild-type (WT) and its derivatives in response to PG and/or pyoluteorin.

To test this possibility, we amended SBW25 carrying pRL-gfp with culture supernatants from Pf-5 or the *ΔphlD* mutant and measured *pltL* expression. Consistent with our hypothesis, culture supernatants from Pf-5, but not the *ΔphlD* mutant, could induce GFP expression of SBW25 (*Figure 3A*). Therefore, we concluded that PG does not directly influence pyoluteorin gene expression. Rather, PG could be converted to a compound(s) that regulates pyoluteorin gene expression.

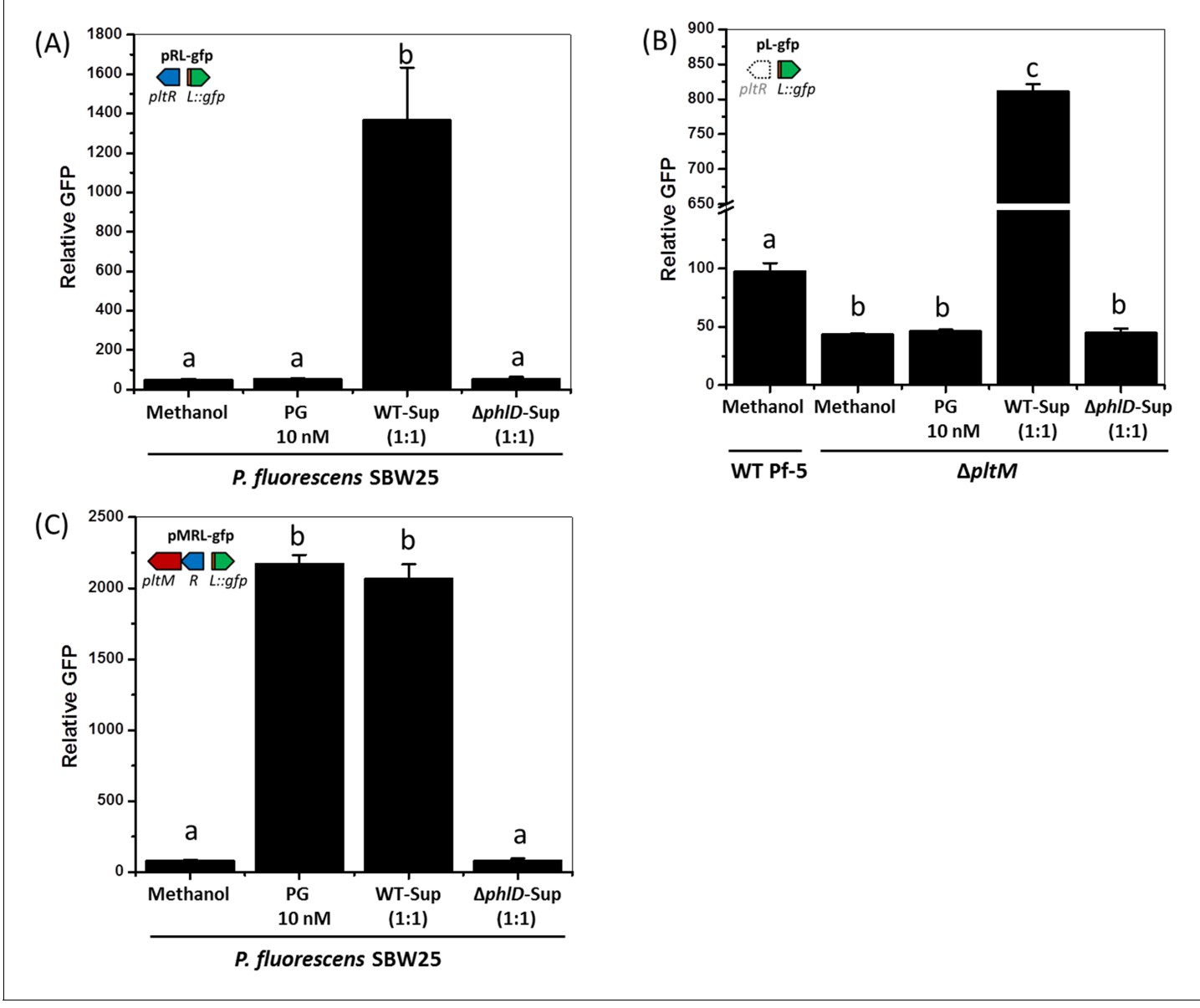

**Figure 3.** Effect of phloroglucinol (PG) and Pf-5 culture supernatants on expression of *pltL::gfp* in *P. fluorescens* SBW25 (A, C), and *P. protegens* Pf-5 (B). (A) Expression of *pltL::gfp* in SBW25 containing pRL-gfp. (B) Expression of *pltL::gfp* in wild-type (WT) Pf-5 and the *ΔphlM* mutant containing pL-gfp. (C) Expression of *pltL::gfp* in SBW25 containing pMRL-gfp. In each experiment, the concentration of PG is 10 nM, and the same volume of methanol was added to the bacterial cultures as a control. The supernatants of bacterial cultures were mixed with the fresh bacterial cultures at a 1:1 vol ratio to test their effects on *pltL::gfp* expression; WT-Sup, the culture supernatants of wild-type Pf-5; *ΔphlD*-Sup, the culture supernatants of a *ΔphlD* mutant. The expression levels of *pltL::gfp* were measured and recorded as relative GFP (fluorescence of GFP divided by $OD_{600}$). Letters above columns indicate treatments significantly different from one another, as determined by ANOVA analysis ($p < 0.05$). Data are means of at least three biological replicates from a representative experiment repeated three times with similar results, and error bars represent the standard deviation of the mean.

The following source data and figure supplements are available for figure 3:

**Source data 1.** Expression of *pltL::gfp* by SBW25 and Pf-5 in response to phloroglucinol (PG) and Pf-5 culture supernatants.

**Figure supplement 1.** Transcriptional profiles of *pltM*, *pltR* and the other genes in the pyoluteorin gene cluster of Pf-5.

**Figure supplement 2.** Production of pyoluteorin, DAPG and MAPG by *P. protegens* Pf-5 wild-type (WT), a *ΔpltM* mutant, and a *pltM+* complemented strain.

*Figure 3 continued on next page*

*Figure 3 continued*

**Figure supplement 2—source data 1.** Concentration of pyoluteorin, MAPG and DAPG (µM) in cultures of Pf-5 wild-type (WT) and its derivatives.

## *pltM* is necessary and sufficient for the PltR-mediated activation of *pltL* expression by PG

We then attempted to identify a mechanism by which Pf-5 could convert PG to the compound(s) that directly induces pyoluteorin gene expression. Within the *plt* gene cluster of Pf-5, *pltM* attracted our attention because of its proximity to *pltR* (*Figure 1C*). *pltM* and *pltR* are predicted to form a bicistronic operon because of a four-base overlap between the 5' end of *pltM* and the 3' terminus of *pltR* (*Paulsen et al., 2005*). Moreover, *pltM* and *pltR* share similar transcriptional profiles, which differ from those of other genes in the *plt* gene cluster (*Figure 3—figure supplement 1*; *Clifford et al., 2016*). PltM is a putative $FADH_2$-dependent halogenase and is required for pyoluteorin production: a *pltM* insertion mutant is deficient in the production of pyoluteorin (*Nowak-Thompson et al., 1999*). The reason why PltM is required is unclear because the *plt* gene cluster encodes another $FADH_2$-dependent halogenase PltA that catalyzes the addition of both chlorines to the pyrrole moiety to form pyoluteorin (*Dorrestein et al., 2005*). Therefore, we hypothesized that PltM is responsible for conversion of PG into the compound(s) that induces the expression of pyoluteorin biosynthetic genes.

To avoid any concerns regarding potential polar effects of the insertion in the previously derived *pltM* mutant (*Nowak-Thompson et al., 1999*), we generated a mutant with an in-frame deletion of *pltM* and confirmed that the *ΔpltM* mutant does not produce pyoluteorin (*Figure 3—figure supplement 2*). Pyoluteorin production by the *ΔpltM* mutant could be complemented by a plasmid-borne copy of *pltM*, confirming that *pltM* has an essential role in pyoluteorin production.

We introduced pL-gfp into the *ΔpltM* mutant and found that the mutant expressed only background levels of GFP, indicating that expression of *pltL* was not induced in this mutant (*Figure 3B*). The GFP expression remained at background levels even when cells were grown in the presence of 10 nM PG. However, GFP expression of the *ΔpltM* mutant was induced by culture supernatants of Pf-5 (*Figure 3B*). As expected, culture supernatants of the *ΔphlD* mutant failed to induce GFP expression of the *ΔpltM* mutant containing pL-gfp. These results supported our hypothesis and indicated that wild-type Pf-5, but not the *ΔpltM* mutant, produces the compound(s) that induces the *pltL* expression.

To test whether *pltM* is sufficient for signaling in SBW25, we made a transcriptional fusion construct pMRL-gfp, which contains *pltM*, *pltR*, and the intergenic region between *pltR* and *pltL* that includes the promoter of *pltL* fused with the promoterless *gfp* (*Figure 1G*). Like SBW25 containing pRL-gfp, SBW25 containing pMRL-gfp had only low, background GFP expression in the non-amended medium, and higher GFP expression level in the presence of culture-supernatants from wild-type Pf-5 (*Figure 3A and C*). While PG failed to induce GFP expression of SBW25 containing pRL-gfp (*Figure 3A*), it induced GFP expression of SBW25 containing pMRL-gfp (*Figure 3C*). These results indicated that *pltM* is sufficient for signaling in SBW25 in the presence of PG and the regulatory gene *pltR*.

## Identification of chlorinated phloroglucinols from Pf-5 cultures

To identify the compound(s) that induce(s) expression of *pltL*, we extracted bacterial metabolites from cultures of the wild-type Pf-5 and the *ΔpltM* mutant, and analyzed the extracts by liquid chromatography–mass spectrometry (LCMS). This analysis revealed the presence of 2-chlorobenzene-1,3,5-triol (PG-Cl) in culture extracts of the wild-type Pf-5 but not the *ΔpltM* mutant (*Figure 4—figure supplement 1*). We also separated the extracts of the wild-type into fractions using semi-preparatory HPLC, and assayed the fractions for signaling activity. This analysis revealed two peaks that induced *pltL* expression in SBW25 (pPL-gfp) (*Figure 4C*), suggesting that Pf-5 produces at least two signaling compounds.

To obtain sufficient levels of signaling molecules for chemical analysis, we constructed a derivative strain of Pf-5 that overproduced the signaling compounds. This overproducing strain is a *ΔgacA* mutant of Pf-5 carrying plasmid pME6010-pltM, which expresses *pltM* from a constitutive

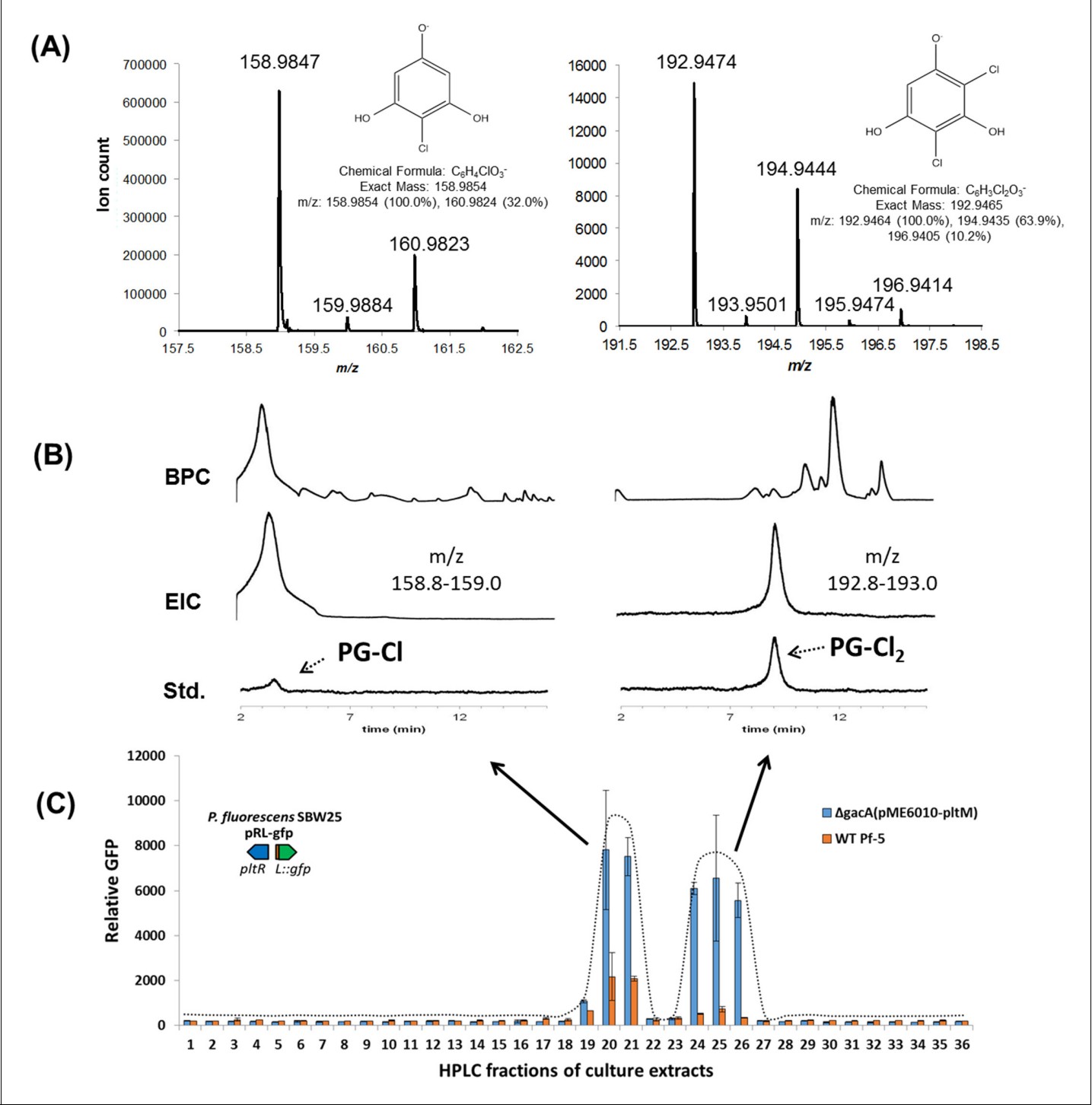

**Figure 4.** Identification of the signaling metabolites from Pf-5 that induce expression of *pltL::gfp* in *P. fluorescens* SBW25 (pRL-gfp). Metabolites were extracted from wild-type (WT) Pf-5, and a phloroglucinol (PG)-fed Δ*gacA* mutant of Pf-5 carrying a plasmid that constitutively expresses *pltM* (pME6010-pltM). The extracts were fractioned into 48 fractions using semi-preparatory HPLC (no observable signal activity was detected from the last 12 fractions, data not shown). The signal activity of the fractions from both strains was tested using strain *P. fluorescens* SBW25 containing pRL-gfp (C). Data are means of three biological replicates, and error bars represent the standard deviation of the mean. (A) Mass spectra of compound eluting at 3.3 min in Fraction 21 (left) and 9 min in Fraction 26 (right) from the Δ*gacA* (pME6010-pltM) culture extracts. (B) Comparisons between the signal compounds identified in Pf-5 cultures to synthesized PG-Cl and PG-Cl₂, which elute at 3.3 and 9 min, respectively, as indicated by the dashed arrows. BPC, base peak chromatogram; EIC, extracted ion chromatogram; Std, synthesized standards. The *m/z* used for EIC analysis of PG-Cl and PG-Cl₂ are also shown.

*Figure 4 continued on next page*

*Figure 4 continued*

The following figure supplement is available for figure 4:

**Figure supplement 1.** LCMS analysis of crude extracts of bacterial metabolites from cultures of wild-type Pf-5 (A) and Δ*pltM* mutant (B).

kanamycin-resistance promoter (Pk) (*Heeb et al., 2000*). The Δ*gacA* mutant does not produce many of the secondary metabolites produced by wild-type Pf-5 (*Hassan et al., 2010*) and thus its use helps to reduce the noise in LCMS analysis. The strain Δ*gacA* (pME6010-pltM) was cultured in a medium amended with PG, and bacterial metabolites from the cultures were extracted and assayed for signal activity using SBW25 (pPL-gfp), as described for the wild-type Pf-5 above. As in the extract of wild-type Pf-5, two peaks in the extract of the Δ*gacA* (pME6010-pltM) strain, spread across six fractions (fractions 19–21, 24–26), showed signal activity (*Figure 4C*). LCMS analysis revealed compounds with protonated ions consistent with PG-Cl in fraction 21 (obs. 158.9847, calc. 158.9854, 4.4 ppm error) and 2,4-dichlorobenzene-1,3,5-triol (PG-Cl$_2$) in fraction 26 (obs. 192.9474, calc. 192.9465, 4.7 ppm error) (*Figure 4A*). The identities of PG-Cl and PG-Cl$_2$ partially purified from the culture extracts were confirmed by comparing them with the synthesized authentic compounds (*Figure 4B*).

PG-Cl, a molecule with previously undescribed biological activity, has been isolated from the red alga *Rhabdoina verticillata* (*Blackman and Matthews, 1982*). To our knowledge, PG-Cl$_2$ has not been previously reported. The structures of these two chlorinated phloroglucinol derivatives are consistent with the expected products of a reaction catalyzed by PltM, an FADH$_2$-dependent halogenase.

## PltM converts PG in vitro into chlorinated phloroglucinols that activate the expression of *pltL*

To determine the catalytic activity of PltM, we characterized the products generated in vitro from PG by PltM. As a putative FADH$_2$-dependent halogenase, PltM is predicted to require a flavin reductase for catalysis (*Nowak-Thompson et al., 1999*), but no putative flavin reductase is encoded in the *plt* gene cluster (*Figure 1C*). Therefore, the previously characterized NAD(P)H-dependent flavin reductase SsuE from *E. coli* (*Keller et al., 2000*) was purified and used here. The results showed that, in the presence of co-factors and flavin reductase, PltM converted PG into a compound(s) that activated *pltL* expression, as determined from GFP expression of SBW25 containing pRL-gfp (*Figure 5*). Furthermore, LCMS analysis showed that both PG-Cl and PG-Cl$_2$ were present in the in vitro reaction containing PltM, but not in the reaction without PltM, based on comparisons with the authentic standards (*Figure 5—figure supplement 1*).

By evaluating a range of compounds related to PG as possible precursors, we also demonstrated that PltM exhibits a high degree of substrate specificity (*Figure 5—figure supplement 2*). These in vitro data, combined with the genetic data above, demonstrate that *pltM* is necessary and sufficient for converting PG into PG-Cl and PG-Cl$_2$, which induce *pltL* expression.

## Chlorinated phloroglucinols induce the PltR-mediated activation of *pltL::gfp* and production of pyoluteorin

We then set out to test the function of PG-Cl or PG-Cl$_2$ in pyoluteorin biosynthesis of Pf-5. Addition of synthesized PG-Cl or PG-Cl$_2$ to cultures induced, in a concentration-dependent manner, expression of *pltL* (*Figure 6A and B*) and pyoluteorin production (*Figure 6C and D*) by the Δ*pltM* mutant of Pf-5. The minimum concentration of PG-Cl$_2$ that induced the expression of *pltL* or production of pyoluteorin was 0.01 μM, which is 100-fold less than that observed for PG-Cl. These results suggest that PG-Cl$_2$ is a more potent signaling compound than PG-Cl. Additionally, there is evidence of toxicity: above 0.1 μM PG-Cl$_2$, growth of *P. protegens* Pf-5 decreased (*Figure 6B*). PG-Cl was also toxic to Pf-5, but only at concentrations of 10 μM or higher. Similarly, PG-Cl and PG-Cl$_2$ showed inhibitory effects on the growth of *P. fluorescens* strain SBW25 (*Figure 6—figure supplement 1*). Induction of *pltL* expression and pyoluteorin production by both PG-Cl and PG-Cl$_2$ required *pltR* (*Figure 6—figure supplements 2* and *3*). This is as expected because PltR is known to be the transcriptional inducer of pyoluteorin biosynthetic genes (*Brodhagen et al., 2004*; *Li et al., 2012*).

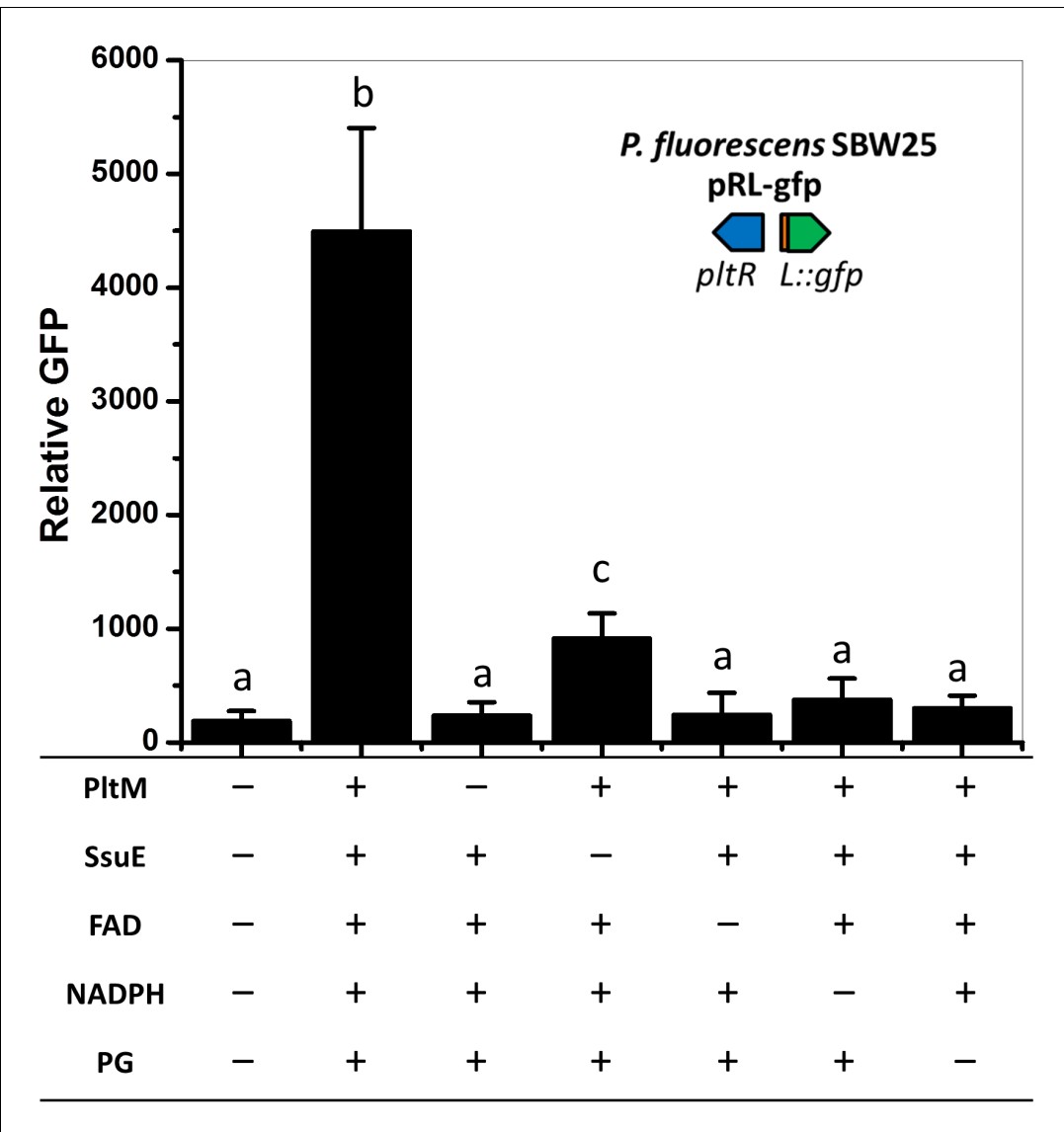

**Figure 5.** Effect of extracts from in vitro catalyzed reactions on expression of *pltL::gfp* in *P. fluorescens* SBW25. PltM was purified from *E. coli* and incubated with a mixture of flavin reductase SsuE, substrate PG, and co-factors FAD and NADPH. The effect of extracted compounds from the reactions was tested using *P. fluorescens* SBW25 containing pRL-gfp. "+" and "−" indicate the presence or absence of the component in the reactions, respectively. The expression level of *pltL::gfp* was measured and recorded as relative GFP (fluorescence of GFP divided by $OD_{600}$). Letters above columns indicate treatments significantly different from one another, as determined by ANOVA analysis ($p < 0.05$). Data are means of at least three biological replicates from a representative experiment repeated twice with similar results, and error bars represent the standard deviation of the mean.

The following source data and figure supplements are available for figure 5:

**Source data 1.** Expression of *pltL::gfp* by SBW25 containing pRL-gfp in response to extracts from in vitro reactions.

**Figure supplement 1.** LCMS analysis of PltM reaction mixtures.

**Figure supplement 2.** Effect of different compounds that have structures similar to phloroglucinol (PG) on the expression of *pltL::gfp* in *P. fluorescens* SBW25.

*Figure 5 continued*

**Figure supplement 2—source data 1.** Expression of *pltL::gfp* by SBW25 containing pMRL-gfp in response to PG and compounds with similar structures to PG.

PG-Cl and PG-Cl$_2$ induced pyoluteorin production concurrently with decreased monoacetylphloroglucinol (MAPG) and DAPG production by the $\Delta pltM$ mutant (*Figure 6C and D*). This result is consistent with the known co-regulation between the pyoluteorin and DAPG biosynthetic pathways (*Brodhagen et al., 2004*).

## PG-Cl and PG-Cl$_2$ function in cell-cell communication and influence antibiosis of Pf-5

DAPG and pyoluteorin are known signals produced by distinct biosynthetic pathways with described roles in intercellular communication of *P. protegens* (*Schnider-Keel et al., 2000*; *Brodhagen et al., 2004*). Several observations suggested that PG-Cl and PG-Cl$_2$ also function as cell-cell communication signals: (1) they are released by the bacterial cell (*Figure 3* and *Figure 4*); (2) they can be taken up by the bacterial cell (*Figure 6*); (3) they regulate gene expression at the transcriptional level and affect the output (antibiotic production) of the cell (*Figure 6*); (4) their production varies with the growth stage of the bacterial cultures (*Figure 7—figure supplement 1*); and (5) their regulatory activities rely on a transcriptional regulator, PltR (*Figure 6—figure supplements 2* and *3*). These results, combined with our previous finding that PG can function as a chemical messenger between cells of *P. protegens* (*Clifford et al., 2016*), collectively support the model depicted in *Figure 7A*, in which PG-Cl and PG-Cl$_2$ can be synthesized from PG by bacterial cells, released into the environment and taken up by neighboring cells to induce expression of pyoluteorin biosynthetic genes and produce pyoluteorin.

We used $\Delta phlA\Delta pltM$ and $\Delta phlA\Delta phlD\Delta pltA$ mutants of Pf-5 to test the model. Both mutants expressed the pyoluteorin biosynthetic gene *pltL* at low levels as indicated by low GFP expression from pL-gfp (*Table 1*, *Figure 7B*). The $\Delta phlA\Delta pltM$ mutant can produce PG but cannot convert it into PG-Cl or PG-Cl$_2$, because of the deletion of *pltM*. Adding PG-Cl or PG-Cl$_2$ to the $\Delta phlA\Delta pltM$ mutant culture led to induced expression of *pltL* (*Table 1*), indicating that this mutant could respond to the signals. The $\Delta phlA\Delta phlD\Delta pltA$ mutant could not produce PG because of the deletion of *phlD*, but still has the capability to convert PG into PG-Cl and PG-Cl$_2$: adding PG to the culture led to increased expression of *pltL* (*Table 1*). Neither mutant could produce pyoluteorin when grown in pure culture, because of the *phlD* and *pltA* mutations in the $\Delta phlA\Delta phlD\Delta pltA$ mutant and the *pltM* mutation in the $\Delta phlA\Delta pltM$ mutant. In contrast, pyoluteorin was produced when the two mutants were grown in co-culture (*Table 1*). Consistently, both mutants showed increased expression of *pltL* in the co-culture compared with the mutants grown in pure culture (*Table 1*, *Figure 7B*). These results support our model (*Figure 7A*) that PG produced by the $\Delta phlA\Delta pltM$ mutant could be converted into PG-Cl and PG-Cl$_2$ by co-cultured cells of the $\Delta phlA\Delta phlD\Delta pltA$ mutant; PG-Cl and PG-Cl$_2$ could then be used by the $\Delta phlA\Delta pltM$ mutant to induce production of pyoluteorin. As expected, the co-culture, but not the pure cultures, of the two mutants inhibited the growth of *E. amylovora* (*Figure 7C*), a plant pathogenic bacterium that is sensitive to pyoluteorin (*Figure 7—figure supplement 2A*). It is worthy to note that neither mutant produces DAPG (*Table 1*) because of the deletion of *phlA*, a structural gene required for DAPG biosynthesis (*Figure 1A and B*), so DAPG production did not obscure the influence of pyoluteorin production on antibiosis.

Collectively, our results support the hypothesis that PG-Cl and PG-Cl$_2$ function in cell-cell communication of *P. protegens* and verify that cells of this bacterium can detect and respond to the chlorinated phloroglucinols produced by neighboring bacterial cells.

## Discussion

The primary contribution of this work was the discovery of a novel mechanism of co-regulation between secondary metabolite biosynthetic pathways (*Figure 8*). This co-regulatory mechanism involves a metabolic intermediate that is transformed through enzymatic activity into a signal regulating gene expression. In contrast to previously described co-regulatory mechanisms involving a

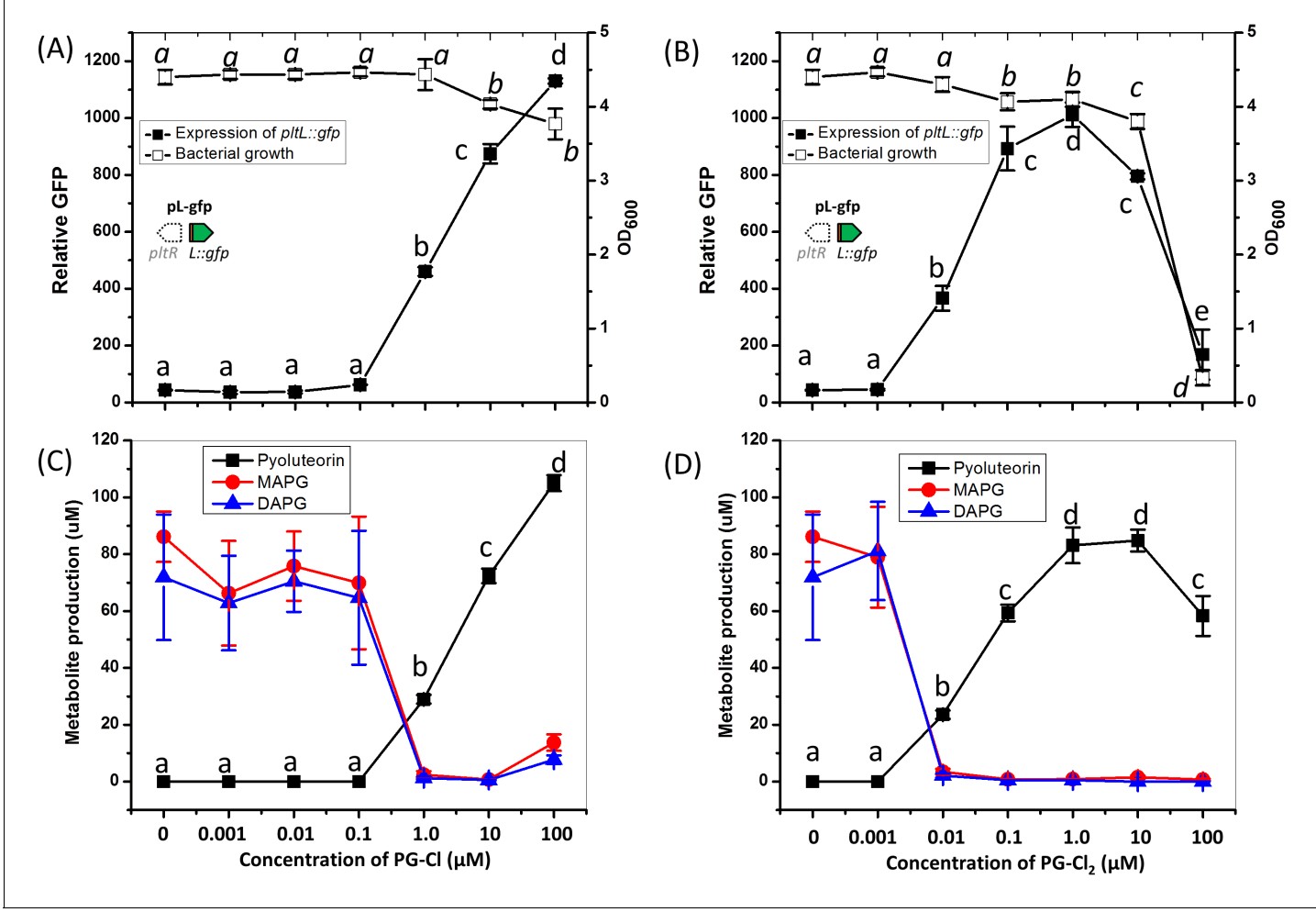

**Figure 6.** Influence of PG-Cl (left) and PG-Cl$_2$ (right) on expression of *pltL::gfp* (**A**, **B**) and production of pyoluteorin (**C**, **D**) in the *ΔpltM* mutant of *P. protegens* Pf-5. The *ΔpltM* mutant was cultured in NBGly amended with increasing concentrations of PG-Cl and PG-Cl$_2$. To test the regulatory effects of PG-Cl and PG-Cl$_2$ on expression of pyoluteorin biosynthetic genes, GFP activity was determined from the *ΔpltM* mutant containing pL-gfp 20 hr after inoculation (**A**, **B**). The OD$_{600}$ value of bacterial cultures (24 hr after inoculation) was measured to show the influence of PG-Cl and PG-Cl$_2$ on the bacterial growth (**A**, **B**). The bacterial cultures were extracted 24 hr after inoculation and production of pyoluteorin, MAPG and DAPG were quantified (**C**, **D**). Letters above the symbols indicate treatments significantly different from one another, as determined by ANOVA analysis ($p < 0.05$). Data are means of at least three biological replicates from a representative experiment repeated twice with similar results, and error bars represent the standard deviation of the mean.

The following source data and figure supplements are available for figure 6:

**Source data 1.** Expression of *pltL::gfp* and production of pyoluteorin by the *ΔpltM* mutant of Pf-5 in response to chlorinated phloroglucinols.

**Figure supplement 1.** Effect of PG-Cl and PG-Cl$_2$ on growth of *P. fluorescens* SBW25.

**Figure supplement 1—source data 1.** Toxicity of PG-Cl and PG-Cl$_2$ to *P. fluorescens* SBW25.

**Figure supplement 2.** Effect of PG-Cl and PG-Cl$_2$ on expression of *pltL::gfp* (**A**) and production of pyoluteorin (**B**) in the *ΔpltMΔphlD* mutant and the *ΔpltMΔpltR* mutant of *P. protegens* Pf-5.

**Figure supplement 2—source data 1.** Expression of *pltL::gfp* and the production of pyoluteorin by Pf-5 wild-type (WT) and its derivatives in response to chlorinated phloroglucinols.

**Figure supplement 3.** Effect of PG-Cl and PG-Cl$_2$ on the expression of *pltL::gfp* in *P. fluorescens* SBW25.

*Figure 6 continued on next page*

*Figure 6 continued*

**Figure supplement 3—source data 1.** Expression of *pltL::gfp* by SBW25 containing pL-gfp or pRL-gfp in response to chlorinated phloroglucinols.

metabolic intermediate (*Vingadassalon et al., 2015*; *Cano-Prieto et al., 2015*), the metabolic intermediate responsible for coordination of DAPG and pyoluteorin production is not shared by the two biosynthetic pathways. Instead, PG is an intermediate in DAPG biosynthesis and a precursor for formation of signaling molecules that activate expression of pyoluteorin biosynthetic genes (*Figure 8*).

The role of antibiotics in cell-cell signaling is now widely recognized (*Yim et al., 2007*), but the results of this study show that intermediates of antibiotic biosynthesis pathways can also function as, or be converted to, signaling molecules. DAPG and pyoluteorin are known to function as intra- and inter-cellular signals (*Brodhagen et al., 2004*; *Maurhofer et al., 2004*; *Powers et al., 2015*). Here we found that the chlorinated phloroglucinols PG-Cl and PG-Cl$_2$ function as intra- and inter-cellular signaling compounds. When added to the culture medium, these compounds can induce expression of pyoluteorin biosynthetic genes in *P. protegens* Pf-5 (*Figure 6*, *Figure 6—figure supplement 2*). By co-culturing mutants of Pf-5 with deficiencies in *phlD* and *pltM*, we also showed that PG, PG-Cl and PG-Cl$_2$ could be released from the producing cells and used by neighboring cells to induce the production of pyoluteorin (*Table 1*, *Figure 7*). These results support and extend those from a previous report showing that PG can serve as an intra- and inter-cellular chemical messenger, influencing the expression of pyoluteorin biosynthetic genes as well as hundreds of other genes with diverse functions in *P. protegens* Pf-5 (*Clifford et al., 2016*). Therefore, it appears that the final products DAPG and pyoluteorin, the biosynthetic intermediate PG, and its derivatives PG-Cl and PG-Cl$_2$ all serve as signals influencing gene expression, the production of antibiotics, and the antimicrobial activity of *P. protegens* (*Figure 8*). The signaling roles of small molecules released as intermediates, derivatives, or end products of antibiotic biosynthesis pathways are likely to be more consequential than previously recognized.

Activation of pyoluteorin biosynthetic genes requires PltR (*Brodhagen et al., 2004*; *Li et al., 2012*) (*Figure 1*), the pyoluteorin pathway-specific positive regulator belonging to the LysR family. LysR regulators typically require a signal for activity, and the signal is commonly an intermediate or product of the regulated gene cluster (*Maddocks and Oyston, 2008*). Pyoluteorin has been proposed to be the signal of PltR (*Li et al., 2012*). However, our data indicate that the signaling effect of pyoluteorin relies on the presence of PG (*Figure 2*), which is converted to PG-Cl and PG-Cl$_2$ (*Figure 3* and *Figure 5—figure supplement 1*). Despite our exhaustive attempts, we were not able to purify the PltR protein to evaluate whether or not PG-Cl or PG-Cl$_2$ binds to PltR in vitro (data not shown). Nevertheless, based on our data that PG-Cl and PG-Cl$_2$ are required for induction of the pyoluteorin biosynthetic genes (*Figure 6*), and *pltR* is required for the signaling action of PG-Cl and PG-Cl$_2$ in both *P. protegens* Pf-5 (*Figure 6—figure supplement 2*) and *P. fluorescens* SBW25 (*Figure 6—figure supplement 2*), we speculate that that PG-Cl and PG-Cl$_2$, rather than pyoluteorin, are signals required for the activity of PltR.

An interesting question is why DAPG and pyoluteorin production are so intricately enmeshed? *P. protegens* can occupy different environments, including the soil, plant surfaces, and insects, and interact with different organisms including bacteria, fungi, protozoa, and insects (*Brodhagen et al., 2004*; *Jousset et al., 2009*; *Kidarsa et al., 2013*; *Henkels et al., 2014*; *Loper et al., 2016*). Both DAPG and pyoluteorin contribute to the antagonism of *P. protegens* against a range of bacteria, fungi, and oomycetes (*Ohmori et al., 1978*; *Keel et al., 1992*) and both facilitate the bacterium's establishment in fungal tissues, such as basidiocarps of the button mushroom *Agaricus* spp. (*Henkels et al., 2014*). However, the sensitivity of a specific organism to these two antibiotics can be different. For example, pyoluteorin is twenty times more toxic than DAPG to the growth of *E. amylovora* (*Figure 7—figure supplement 2*). Furthermore, these two compounds also have distinct ecological roles. For example, DAPG but not pyoluteorin exhibits toxicity to the amoeba *Acanthamoeba castellanii* (*Jousset and Bonkowski, 2010*). Therefore, natural situations may exist in which it could be more advantageous to produce one compound versus the other. Additionally, antibiotic biosynthesis is associated with metabolic costs. The coordinated production of pyoluteorin and DAPG may be a mechanism for the bacterium to balance metabolic cost against the benefits

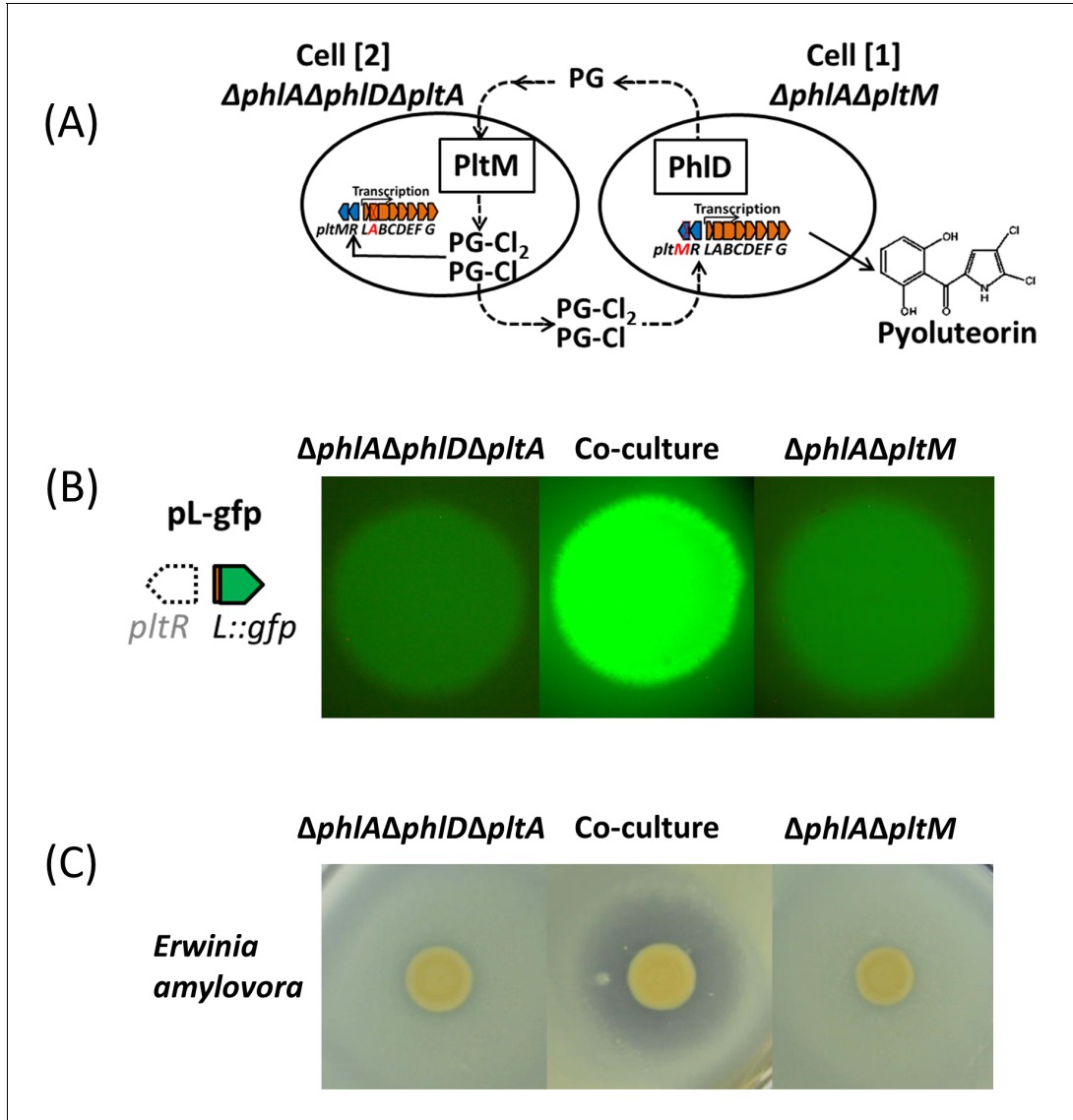

**Figure 7.** PG-Cl and PG-Cl$_2$ function as cell-cell communication signals of *P. protegens* Pf-5. (**A**) Proposed cell-cell communication model. Cell [1] (*ΔphlAΔpltM*) produces phloroglucinol (PG) and releases it into the environment. PG is taken up by cell [2] (*ΔphlAΔphlDΔpltA*) and converted into PG-Cl and PG-Cl$_2$, which are secreted into the environment. PG-Cl and PG-Cl$_2$ are taken up by cell [1] and induce expression of pyoluteorin biosynthetic genes and production of pyoluteorin by cell [1]. The *plt* gene cluster is shown. Red font indicates that the gene was deleted. (**B**) GFP expression from the pure- and co-culture of the *ΔphlAΔpltM* mutant containing pL-gfp and the *ΔphlAΔphlDΔpltA* mutant containing pL-gfp on NAGly plate. (**C**) Antibiosis of the pure- and co-culture of the *ΔphlAΔpltM* mutant and *ΔphlAΔphlDΔpltA* mutant against growth of *Erwinia amylovora* on a NAGly plate. The co-culture was prepared by mixing the two mutants 1:1 and spotting on the plate. The experiment was repeated at least twice, with similar results.

The following source data and figure supplements are available for figure 7:

**Figure supplement 1.** Levels of signaling compounds in cultures of *P. protegens* Pf-5 of different ages.

**Figure supplement 1—source data 1.** Expression of *pltL::gfp* by SBW25 in response to extracts of wild-type Pf-5 cultures.

**Figure supplement 2.** Toxicity of pyoluteorin (**A**) and DAPG (**B**) to the plant pathogen *Erwinia amylovora*.

*Figure 7 continued on next page*

**Table 1.** Concentrations of DAPG, pyoluteorin, and expression of *pltL::gfp* in pure cultures and co-culture of the *ΔphlAΔpltM* mutant and the *ΔphlAΔphlDΔpltA* mutant of *P. protegens* Pf-5.

| Product/expression* | Amendment[†] | Bacterial cultures[‡] | | |
|---|---|---|---|---|
| | | *ΔphlAΔpltM* | *ΔphlAΔpltM + ΔphlAΔphlDΔpltA* | *ΔphlAΔphlDΔpltA* |
| Pyoluteorin | None | BD | 6.2 ± 1.8 nmol/g | BD |
| DAPG | None | BD | BD | BD |
| Relative GFP | None | 34.6 ± 2.3 | 137.4 ± 3.4[§]; 214.7 ± 11.2[¶] | 27.6 ± 1.9 |
| | PG (10 nM) | 37.6 ± 3.1 | NT | 133.3 ± 6.4 |
| | PG-Cl (1 µM) | 261.0 ± 13.3 | NT | 308.6 ± 21.8 |
| | PG-Cl$_2$ (10 nM) | 128.2 ± 10.7 | NT | 268.4 ± 16.2 |

*The mutants were cultured for 3 d on a NAGly plate to determine production of secondary metabolites, and for 20 hr in NBGly broth to determine expression of *pltL::gfp* from the reporter construct pL-gfp (**Figure 1E**). The co-culture was prepared by mixing the two mutants at a 1:1 ratio.

[†]PG, PG-Cl or PG-Cl$_2$ was amended in the mutant cultures to test their influences on the expression of *pltL::gfp*. The concentrations of the tested compounds are shown in parenthesis.

[‡]Secondary metabolites were extracted from the agar plate and the concentrations of DAPG and pyoluteorin were determined by HPLC. Expression of *pltL::gfp* was monitored and recorded as relative GFP (fluorescence of GFP divided by OD$_{600}$). [§]The *ΔphlAΔpltM* mutant, but not the *ΔphlAΔphlDΔpltA* mutant, contains pL-gfp in the co-culture. [¶]The *ΔphlAΔphlDΔpltA* mutant, but not the *ΔphlAΔpltM* mutant, contains pL-gfp in the co-culture. Data are presented as mean ± standard derivation. Mean values are calculated from three biological replicates. BD = below detection. NT = not tested.

conferred by production of the two antibiotics. In short, metabolic co-regulation may enable bacteria to activate the production of one antibiotic to combat certain competitors or predators or to occupy specific habitats, while reducing the metabolic burden on the cell by repressing the production of a second compound.

In summary, this study demonstrated that PG, an intermediate in the DAPG biosynthetic pathway, is converted into PG-Cl and PG-Cl$_2$, which serve as signals activating expression of pyoluteorin biosynthetic genes. We also showed that PG-Cl and PG-Cl$_2$ can be released from and sensed by bacteria to regulate antibiosis against a bacterial pathogen, indicating that these metabolites function as cell-cell communication signals. Many questions remain related to the co-regulation of the DAPG and pyoluteorin biosynthesis pathways. Membrane receptors recognizing the various signaling compounds (DAPG, pyoluteorin, PG, PG-Cl and PG-Cl$_2$) have not been identified, but are likely needed to facilitate uptake of these molecules across the bacterial membrane. PG-Cl and PG-Cl$_2$ are now known to be responsible for the previously described activation of the pyoluteorin biosynthesis by nanomolar concentrations of PG, but these compounds do not appear to mediate the repression of pyoluteorin production by higher (micromolar) concentrations of PG (**Kidarsa et al., 2011**). The mechanism(s) involved in inhibition of pyoluteorin production by PG remain unknown. Here, we report a novel mechanism of co-regulation that is responsible for only one aspect of the complex and intricate co-regulation of two secondary metabolites in *P. protegens*. Our results highlight the diversity of mechanisms involved in metabolic co-regulation in bacteria and provide an unprecedented example of co-regulation between separate biosynthetic pathways in which the intermediate of one pathway functions as a precursor of the signal(s) activating the biosynthesis of a second pathway.

## Materials and methods

### Strains, plasmids, primers, culture conditions and chemical compounds

The bacterial strains and plasmids used in this study are listed in *Table 2*. The sequences of oligonucleotides used in this study are listed in *Table 3*.

*P. protegens* Pf-5 and derivatives, *P. fluorescens* SBW25, and *Erwinia amylovora* were cultured at 27°C on King's Medium B agar, Nutrient Agar (Becton, Dickinson and Company, Sparks, MD) supplemented with 1% glycerol (NAGly) or Nutrient Broth (Becton, Dickenson and Company) supplemented with 1% glycerol (NBGly). Liquid cultures were grown with shaking at 200 r.p.m.

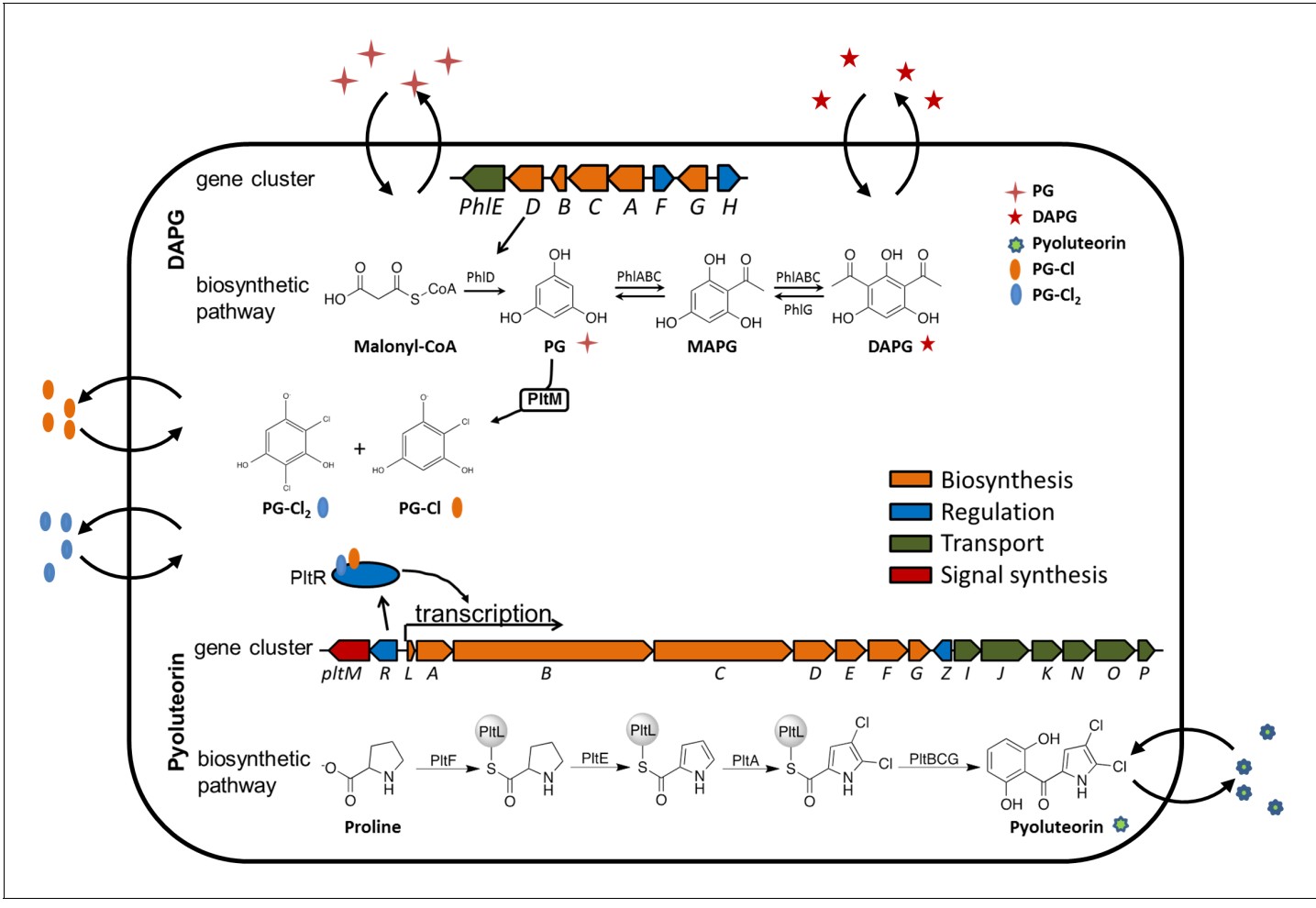

**Figure 8.** Proposed model of metabolic co-regulation between DAPG and pyoluteorin biosynthetic pathways in *P. protegens*. Phloroglucinol (PG) is synthesized by the type III polyketide synthase PhlD in the DAPG biosynthetic pathway, and processed into PG-Cl and PG-Cl$_2$ by the halogenase PltM encoded in the pyoluteorin gene cluster. The PG-Cl and PG-Cl$_2$ putatively bind to regulator PltR and induce transcription of pyoluteorin biosynthetic genes and prompt production of pyoluteorin. In these two pathways, the final products (DAPG and pyoluteorin), the biosynthesis intermediate (PG) and its derivatives (PG-CL, PG-Cl$_2$) can be released and used by bacteria and function as cell-cell communication signals.

PG was purchased from Sigma-Aldrich (St. Louis, MO, USA), MAPG was purchased from Sigma Aldrich Chemie GmbH (Schnelldorf, Germany), and DAPG was purchased from Toronto Research Chemicals (North York, Canada). Chlorinated phloroglucinols PG-Cl, PG-Cl$_2$ and PG-Cl$_3$ were purchased from Synthon-Lab (St. Petersburg, Russia). All compounds were >99% pure, as determined by HPLC, and were solubilized in methanol.

## Construction of Pf-5 mutants and derivative strains

The *ΔpltAΔphlD* mutant was made by sequentially deleting *phlD* and *pltA* genes from the chromosome of wild-type *P. protegens* Pf-5. To delete *phlD* from the chromosome of Pf-5, a *phlD* deletion construct made previously (*Kidarsa et al., 2011*) was transferred into strain *E. coli* S17-1 and used to delete 888 bp from the *phlD* gene in the chromosome of Pf-5 using a previously described protocol (*Kidarsa et al., 2011*). To further delete *pltA*, a *pltA* deletion construct made previously (*Henkels et al., 2014*) was used to delete 275 bp from the *pltA* gene in the chromosome of Pf-5. The deletion of *phlD* and *pltA* was confirmed by PCR and subsequent DNA sequencing.

To delete *pltR* from the chromosome of Pf-5, two DNA fragments flanking the *pltR* gene were PCR amplified using the oligonucleotide pair pltR UpF-Xba/pltR UpR and pltR DnR-Xba/pltR DnF (*Table 3*). These two fragments were fused together by PCR and digested using XbaI to generate a

**Table 2.** Bacterial strains and plasmids used in this study.

| Strains and plasmids | Genotype and relevant characteristics* | Reference or source |
|---|---|---|
| **Strains** | | |
| *P. protegens* | | |
| LK099 | Wild-type Pf-5 | (*Howell and Stipanovic, 1979*) |
| JL4928 | $\Delta pltA\Delta phlD$ | This study |
| LK269 | $\Delta pltA\Delta phlD\Delta pltR$ | This study |
| LK270 | $\Delta pltM$ | This study |
| LK024 | $\Delta phlD$ | (*Kidarsa et al., 2011*) |
| LK293 | $\Delta pltM\Delta pltR$ | This study |
| LK397 | $\Delta phlD\Delta pltM$ | This study |
| LK096 | $\Delta pltA\Delta phlA$ | (*Henkels et al., 2014*) |
| LK403 | $\Delta phlA\Delta pltM$ | This study |
| LK413 | $\Delta phlA\Delta phlD\Delta pltA$ | This study |
| *P. fluorescens* SBW25 | SBW25 is a member of the *P. fluorescens* group but does not contain the *phl* or *plt* gene clusters | (*Silby et al., 2009*) |
| *E. coli* | | |
| BL21(DE3) | *ompT hsd*S$_B$ (r$_B^-$m$_B^-$) *gal dcm* | NEB |
| S17-1 | *recA pro hsdR*$^-$M$^+$ RP4 2-Tc::Mu-Km::*Tn7* Sm$^r$ Tp$^r$ | (*Simon et al., 1983*) |
| **Plasmids** | | |
| pEX18Tc | Gene replacement vector with MCS from pUC18, *sacB*$^+$ Tc$^r$ | (*Hoang et al., 1998*) |
| p18Tc-$\Delta$pltM | pEX18Tc containing *pltM* with an internal deletion of 1345 bp | This study |
| pPROBE-NT | pBBR1 containing promoterless *gfp*, Km$^r$ | (*Miller et al., 2000*) |
| pRL-gfp | pPROBE-NT containing *pltR* as well as the intergenic region between *pltR* and *pltL* including the promoter of *pltL* fused with a promoterless *gfp* | This study |
| pMRL-gfp | pPROBE-NT containing *pltM* and *pltR* as well as the intergenic region between *pltR* and *pltL* including the promoter of *pltL* fused with a promoterless *gfp* | This study |
| pL-gfp | Called ppltL-gfp previously. Contains the intergenic region between *pltR* and *pltL* including the promoter of *pltL* fused with a promoterless *gfp* | (*Yan et al., 2016*) |
| pME6010 | pACYC177-pVS1 shuttle vector, Tc$^r$ | (*Heeb et al., 2000*) |
| pME6010-pltM | pME6010 containing *pltM*, which is expressed from the constitutive promoter *Pk* | This study |
| pET28a(+) | Protein overexpression vector | Novagen |
| pET28a-PltM | pET28a containing *pltM*, used for overexpression of the PltM protein | This study |
| pET28a-SsuE | pET28a containing *ssuE*, used for overexpression of the SsuE protein | This study |

*Sm$^r$, streptomycin resistance; Tp$^r$, trimethoprim resistance; Tc$^r$, tetracycline resistance; Km$^r$, kanamycin resistance.

1325 bp DNA fragment containing *pltR* with a 925 bp internal deletion. This DNA fragment was ligated to pEX18Tc to create construct p18Tc-$\Delta$pltR. This deletion construct was introduced into the $\Delta pltA\Delta phlD$ mutant to make the $\Delta pltA\Delta phlD\Delta pltR$ triple mutant. The deletion of *pltR* was confirmed by PCR and subsequent DNA sequencing.

To delete *pltM* from the chromosome of Pf-5, two DNA fragments flanking the *pltM* gene were PCR amplified using the oligonucleotide pair pltMRL-F1/pltM-up-ovlp and pltM-dn-ovlp/pltRf. These two fragments were fused together by PCR and digested using EcoRI and KpnI to generate a 1120 bp DNA fragment containing *pltM* with a 1345 bp internal deletion. This DNA fragment was ligated to pEX18Tc to create construct p18Tc-$\Delta$pltM. The construct p18Tc-$\Delta$pltM was transferred into strain *E. coli* S17-1 and used to delete 1345 bp from the *pltM* gene in the chromosome of Pf-5. The deletion of *pltM* from Pf-5 was confirmed by PCR and subsequent DNA sequencing.

**Table 3.** Sequences of oligonucleotides used in this study

| Oligonucleotides | DNA sequences* |
|---|---|
| pltR UpF-Xba | GAGAGG<u>TCTAGA</u>AGTGGAGTCTGGTCATCAAG |
| pltR UpR | GCTCTTGTCTTGTAGTCTTCCTGTTTCGGAGAA |
| pltR DnF | AACAGGAAGACTACAAGACAAGAGCCAGACATC |
| pltR DnR-Xba | GGTGTG<u>TCTAGA</u>GTGAAGAATGAGCAGGTGTC |
| pltM-F1 | AT<u>GGTACC</u>CTGCATTTCGATACCGC |
| pltM-R1 | ATA<u>GAATTC</u>GGCGGACGGGTGGATC |
| pltMRL-F1 | AGCGATTTCGATCTTCATCCCC |
| pltM-up-ovlp | GAAGCGGCGACACCGGATCCAATCGTTTCATCC |
| pltM-dn-ovlp | GGATGAAACGATTGGATCCGGTGTCGCCGCTTC |
| pltRf | AT<u>GGTACC</u>AAGGATTTAGGAATGAAGGC |
| pltM 5'primer | GTCATA<u>CATATG</u>AATCAGTACGACGTCATTATC |
| pltM 3' primer | CAGTGC<u>CTCGAG</u>TCAGACTTTGAGGATGAAACG |
| ssuE 5' primer | GGAGAG<u>CATATG</u>CGTGTCATCACC |
| ssuE 3' primer | GTA<u>AAGCTT</u>TTACGCATGGGCATT |
| pltM 3' primer3 | ATA<u>GGATCC</u>GGCCCCGGGCATCACTCAG |
| pltRr | TAT<u>AAGCTT</u>TCAGCCCGGACTTCGCGAGG |
| gfp-plt-r1 | AT<u>GGTACC</u>ATAGACGTACGCTCCTGC |

*Restriction sites used for cloning are underlined in oligonucleotides.

---

To complement the *ΔpltM* mutant, a 1661 bp DNA fragment containing *pltM* was PCR amplified using oligonucleotides pltM-F1 and pltM-R1. This PCR product was digested by KpnI and EcoRI and ligated downstream of the constitutively expressing promoter (P*k*) of the kanamycin resistance gene in pME6010 to make the complementation construct pME6010-pltM. The plasmid pME6010-pltM was transferred into Pf-5 derivatives by bi-parental mating (*Kidarsa et al., 2011*).

## Construction of protein overexpression constructs

For heterologous expression in *Escherichia coli*, the *pltM* coding sequence was PCR amplified using oligonucleotides pltM 5'primer and pltM 3'primer. The PCR product was digested with NdeI and XhoI, and ligated to the expression vector pET28a(+) to create the expression construct pET28a-PltM, in which *pltM* was expressed with a N-terminal fused 6X His Tag. Similarly, for SsuE, the *ssuE* coding sequence was PCR amplified using *E. coli* K12 genomic DNA as a template, and oligonucleotides ssuE 5'primer and ssuE 3'primer. The PCR product was digested with NdeI and HindIII and ligated to the expression vector pET28a(+) to create the expression construct pET28a-SsuE, in which SsuE was expressed with a N-terminal fused 6X His Tag.

## Construction of transcriptional reporter constructs

For reporter constructs, three variants of *pltL*::GFP were constructed. pL-gfp contains the promoter of *pltL* fused to a promoterless *gfp* (*Yan et al., 2016*). The construct pRL-gfp includes the entire *pltR* coding sequence and the predicted promoter sequence of *pltL*. To make this construct, a 1600 bp DNA fragment was PCR amplified using oligonucleotides pltRr and gfp-plt-r1. The resultant PCR fragment was digested with HindIII and KpnI and ligated to pPROBE-NT to create the construct pRL-gfp. The construct pMRL-gfp contains *pltM* and *pltR* and the promoter region of *pltL* fused to a promoterless *gfp*. To make this construct, a 3050 bp DNA fragment was PCR amplified using pltM 3'primer3 and gfp-plt-r1. The resultant PCR fragment was digested with BamHI and KpnI and ligated to pPROBE-NT to create the construct pMRL-gfp.

## GFP quantification

Strains *P. protegens* Pf-5 and *P. fluorescens* SBW25 containing *gfp*-based reporter plasmids were cultured overnight in NBGly at 27°C with shaking. The cells were washed once with fresh NBGly and used to inoculate 5 ml NBGly to an optical density of 600 nm ($OD_{600}$) of 0.01. Five microliters of purified compounds at various concentrations (pyoluteorin, PG, and chlorinated derivatives of PG) or extracts (described below) were added to the 5 ml cultures. The cultures were incubated for 20 hr at 27°C with shaking. The $OD_{600}$ was measured to monitor growth of the bacteria. The green fluorescence of bacteria was monitored using a 96-well plate reader (Tecan Infinite 200Pro, Männedorf, Switzerland) by measuring emission at 535 nm with an excitation at 485 nm. For each measurement, the green fluorescence value was divided by the corresponding $OD_{600}$ to determine the relative GFP level. The green fluorescence emitted by cells of wild-type strains without *gfp* reporter constructs was determined and used for background correction.

## Protein purification and in vitro activity assay

The expression plasmids pET28a-PltM and pET28a-SsuE were transformed into *E. coli* BL21 (DE3). Cells were grown in LB broth, supplemented with 50 mg/ml kanamycin, at 37°C to an $OD_{600}$ of 0.5, induced with 0.4 mM IPTG, and grown for an additional 12 hr at 20°C. The cells were harvested and suspended in Tris-NaCl buffer (20 mM Tris pH 7.4/200 mM NaCl), and lysed by sonication. The His-tagged proteins were purified using the Ni-NTA Purification System (Invitrogen) under native purification conditions. Purified PltM and SsuE were dialyzed in Tris-NaCl-Glycerol buffer [20 mM Tris pH 7.4/200 mM NaCl/10% (vol/vol) glycerol]. The protein concentrations were determined using a *DC* Protein Assay (Bio-Rad) and proteins were stored at −80°C.

To test the catalytic activity of PltM, 20 µl of 56 µM PltM, 50 µl of 21 µM SsuE, 4 µl of 10 mM FAD, 10 µl of 100 mM NADPH, and 10 µl of 10 mM PG were added to 106 µl of Tris-NaCl buffer (20 mM Tris pH 7.4/200 mM NaCl). Five additional reactions were also set up, with one of the five components replaced by an equivalent volume of the Tris-NaCl buffer in each reaction. The mixtures were incubated for 3 hr at 25°C and extracted three times with ethyl acetate. The ethyl acetate extracts were dried under vacuum at room temperature and dissolved in 200 µl methanol. A volume of 5 µl from each methanol solution was added to *P. fluorescens* SBW25 containing the reporter plasmid pRL-gfp to determine the signaling activity of each reaction product. For LCMS analysis, the reactions were concentrated to dryness and dissolved in 100 µl methanol and 10 µl were analyzed as described below.

## Quantification of MAPG, DAPG, and pyoluteorin

Pf-5 and derivative strains were inoculated at a starting $OD_{600}$ of 0.01 in 5 ml NBGly and cultured for 24 hr. Four milliliters of the culture supernatant were extracted twice with ethyl acetate, then subsequently dried under vacuum, and suspended in methanol. The culture extracts were analyzed by high-performance liquid chromatography (HPLC) to quantify production of MAPG, DAPG, and pyoluteorin as described below.

To quantify the production of MAPG, DAPG and pyoluteorin by Pf-5 and derivatives, HPLC analyses were accomplished using an Agilent 1100 HPLC instrument, which consisted of a quaternary pump, vacuum degasser, autosampler, column thermostat (set to 30°C), and diode array detector. Separation was achieved using a Luna C18 column (4.6 × 150 mm, 5 µm, Phenomenex, Torrance, CA) with a flow rate of 1 ml/min where line A was water + 0.1% (vol/vol) formic acid, and line B was acetonitrile + 0.1% (vol/vol) formic acid with the following program. The column was pre-equilibrated in 90% A/10% B, and upon injection this composition was held for 2 min. The composition of the mobile phase was then changed to 0% A/100% B over 28 min using a linear gradient. This composition was held for 6 min then changed to 90% A/10% B over 2 min. The column was equilibrated in 90% A/10% B for 6 min prior to the next injection. Under these chromatographic conditions, MAPG eluted at 12.1 min, pyoluteorin eluted at 15.1 min, and DAPG eluted at 18.1 min. The HPLC was operated with and data viewed withChemStation (version B.04.03, Agilent, Santa Clara, CA). Quantification was performed by integrating the area under the curve at 300 nm and comparing with a standard curve prepared by injection of purified pyoluteorin, MAPG, and DAPG. Data were processed with GraphPad Prism (GraphPad Software, San Diego, CA).

## Identification of chlorinated phloroglucinols

For LCMS analysis of crude culture extracts of the wild-type Pf-5 and the $\Delta pltM$ mutant, these two strains were grown for 20 hr in 500 ml NBGly at 27°C with shaking. The supernatants of the cultures were extracted twice with ethyl acetate (250 ml). The extracts were concentrated in vacuo to dryness. The dried extracts were dissolved in 50 µl methanol and then diluted to 400 µl with water. The large amount of precipitate formed was removed by centrifugation (15,000 x $g$, 60 min, 10°C), and 10 µl of the clear supernatant were analyzed by LCMS as described below.

LCMS analysis was performed using a Agilent 1260 HPLC (consisting of degasser, quaternary pump, autosampler, and diode array detector) upstream of a 6230 ESI-TOF operated in negative ionization mode with the following parameters: Mass range, 100–3200 m/z in profile mode; Gas temperature, 350°C; Drying gas, 10 L/min; Nebulizer, 40 psig; Capillary voltage, 3500 V; Fragmentor, 100 V; Skimmer, 65 V; OCT 1 RF Vpp, 750 V; Acquisition rate, five spectra/sec; Time, 200 ms/spectrum; Transients/spectrum, 1902). The synthesized authentic compounds PG-Cl, PG-Cl$_2$ and PG-Cl$_3$ were used as standards for comparisons in the analysis. Separation was achieved with an Extend C18 column (2 × 100 mm, 3 µm, Agilent) at a flow rate of 0.2 ml/min. The column was pre-equilibrated in 98% A/2% B, where A was water + 0.1% (vol/vol) formic acid and B was acetonitrile + 0.1% (vol/vol) formic acid. This composition was held for 1 min and then changed to 100% B over 30 min using a linear gradient. After that, the composition was held for an additional 12 min and the composition was returned to 98% A/2% B over 2 min. The column was equilibrated at 98% A/2% B for 12 min prior to injection of the next sample. The LCMS was operated with and data were viewed using MassHunter (version B.04.03, Agilent, Santa Clara, CA).

Culture extracts from the wild-type Pf-5 were also fractioned by semi-preparatory HPLC and signal activity of the fractions was tested by the reporter strain *P. fluorescens* SBW25 containing pRL-gfp. The wild-type Pf-5 was cultured for 20 hr in 500 ml NBGly at 27°C with shaking before being extracted twice with ethyl acetate (250 ml). The dried extracts were dissolved in 1 ml methanol and mixed with 1 ml water. The concentrated extracts were separated by semi-preparatory HPLC using a PuriFlash Pf450 (Interchim Inc., Los Angeles, CA). Separation was achieved using a Purisphere C18 column (10 × 150 mm, 10 µm, Interchim Inc.) at a flow rate of 2 ml/min. The column was pre-equilibrated in 90% A/10% B, where A was water and B was methanol. This composition was held for 2 min and then increased to 100% B over 40 min using a linear gradient and pure methanol was flown over the column for an additional 9 min. The effluent from the pump was collected directly into a 96-deep well plate in 1 min time slices after discarding the first 3 min to waste. Five microliters from each well were tested using SBW25 (pRL-gfp) and fluorescence was observed as described above.

To obtain sufficient levels of signaling molecules for LCMS analysis, the $\Delta gacA$ mutant of Pf-5 containing pME6010-pltM was cultured in three flasks, each containing 1000 ml NBGly amended with 12.5 µg/ml PG. The cultures were incubated for 20 hr at 27°C with shaking before being extracted twice with ethyl acetate (300 ml). Extracts were concentrated in vacuo to dryness. The dried extracts from the three replicate cultures were combined and dissolved in 1.5 ml methanol. A portion (200 µl) was diluted with 1.8 ml of water. The concentrated extracts were fractioned by semi-preparatory HPLC and the signal activity of the fractions were tested using SBW25 (pRL-gfp) as described above. The fractions with the highest signal activity were concentrated to dryness separately and analyzed by LCMS along with synthesized standards including PG-Cl, PG-Cl$_2$ and PG-Cl$_3$ as described above.

## Antibiosis assay

Antibiosis of Pf-5 and derivatives against the plant pathogen *E. amylovora* was determined on NAGly. *E. amylovora* was cultured overnight in 5 ml NBGly at 27°C with shaking, and 80 µl of the cultures were inoculated into 8 ml melted NAGly (45°C), and poured over solidified NAGly in a Petri plate. This two-layer plate was air-dried with the lid open for 1 hr to solidify the agar and remove extra moisture. The $\Delta phlA\Delta pltM$ mutant and the $\Delta phlA\Delta phlD\Delta pltA$ mutant were cultured overnight in 5 ml NBGly at 27°C with shaking. The cells were washed once with fresh NBGly and used to make a bacterial suspension with an OD$_{600}$ of 0.05. For co-culture, cells of the two mutants were combined in a 1:1 ratio for a total OD$_{600}$ of 0.05. Two microliters of the bacterial suspensions were used to inoculate three two-layer plates and incubated for 3 d at 27°C. The agar medium with the Pf-5 strains in the center was cut from the plate and concentrations of DAPG and pyoluteorin in the agar were

quantified as follows. Bacterial cells were carefully removed from the agar using Kimwipe. The agar was weighed, sliced into small pieces and immersed for 30 min in sterilized water before extracting twice with ethyl acetate. The extracts were dried and the residue was dissolved in 20 μl methanol and analyzed by HPLC as described above.

### Statistical analysis

Statistical analyses were performed by ANOVA and Tukey multiple comparison using SPSS statistics 20 (SPSS Inc., Chicago, IL). All replicates in this study were biological replicates. Sample sizes are indicated in the figure legends. All data are presented as mean ± standard derivation.

## Acknowledgements

We are grateful to Marcella Henkels, Brenda Shaffer, and Max Kohen for their assistance; Jennifer Clifford, Teresa Kidarsa, Steven Lindow, and Virginia Stockwell for helpful discussions; and the Center for Genomics Research and Biocomputing at Oregon State University for sequencing services.

## Additional information

### Funding

| Funder | Grant reference number | Author |
| --- | --- | --- |
| National Institute of Food and Agriculture | NRI 2011-67019-30192 | Jeff H Chang Joyce E Loper |
| Oregon State University | Research startup funds | Benjamin Philmus |

The funders had no role in study design, data collection and interpretation, or the decision to submit the work for publication.

### Author contributions

QY, Conceptualization, Formal analysis, Investigation, Writing—original draft, Writing—review and editing; BP, Funding acquisition, Investigation, Writing—review and editing; JHC, Supervision, Funding acquisition, Writing—review and editing; JEL, Conceptualization, Supervision, Funding acquisition, Project administration, Writing—review and editing

### Author ORCIDs

Qing Yan, http://orcid.org/0000-0002-7830-9126
Joyce E Loper, http://orcid.org/0000-0003-3501-5969

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
