## [Decision Letter]

Thank you for submitting your article "Novel mechanism of metabolic crosstalk coordinates the biosynthesis of secondary metabolites in *Pseudomonas protegens*" for consideration by *eLife*. Your article has been favorably evaluated by Richard Losick (Senior Editor) and three reviewers, one of whom, Jon Clardy (Reviewer #1), is a member of our Board of Reviewing Editors. The following individuals involved in review of your submission have agreed to reveal their identity: Pierre Stallforth (Reviewer #2) and Sarah O'Connor (Reviewer #3).

The reviewers have discussed the reviews with one another and the Reviewing Editor has drafted this decision to help you prepare a revised submission.

Summary:

This manuscript reports an interesting case in which two antibiotic biosynthesis pathways in *Pseudomonas protegens* are linked through having an intermediate for one be modified by an enzyme in the second to produce a third molecule that up-regulates the biosynthesis of the second antibiotic. The two antibiotics – diacylphloroglucinol (DAPG) and pyoluteorin (PLT) – are well known and have distinct if overlapping activities. This study reports that phloroglucinol (PG), a precursor of DAPG, is modified by a chlorinase in the PLT biosynthetic pathway to produce chlorinated versions of PG, PG-Cl and PG-Cl_2_. The chlorinated PGs function as positive regulators of PLT biosynthesis, and are themselves antibiotics. The latter finding is not surprising, as they resemble carbolic acid – an antibiotic going back to Joseph Lister. The study is well done, and the results are unexpected.

Essential revisions:

The reviewers were uniformly enthusiastic about publishing your article after some revision. Major issues that need to be addressed are summarized below, along with the individual review of reviewer #3.

1) Enhanced mechanistic analysis of the co-regulation including the difference/similarity of phlD and pltD, apparently redundant enzymes for PG production. A mechanistic analysis would be facilitated by using "co-regulation", which is the much more informative description used in the impact statement, rather than "crosstalk", which following its more common definition of "components of different signal transduction pathways affecting one another", is inappropriate.

2) The structural analysis of PG-Cl and PG-CL_2_ are not up to the usually accepted standards for establishing the structures of new molecules. It should be relatively easy to provide the additional data.

3) Suggest that the title be modified to: Metabolic co-regulation coordinates the biosynthesis of secondary metabolites in *Pseudomonas* protegens.

*Reviewer #3:*

This manuscript demonstrates how a *Pseudomonas* species uses crosstalk between the two natural product pathways, DAPG and pyoluteorin, to control the regulation of these natural product biosynthesis pathways. While metabolic crosstalk has been previously observed on many other occasions, the authors convincingly demonstrate how this is an unusual and unprecedented mechanism. The authors use an assay based on the transcription of a reporter protein to identify the two signaling molecules responsible for the crosstalk, in this case two chlorinated derivatives of a biosynthetic intermediate of the DAPG biosynthetic pathway. Overall, this is an elegant study that shows an unusual mechanism of crosstalk. I believe that it is appropriate for the broad readership of *eLife*.

I have the following comments below:

In Figure 5 only the mass of the products is shown, which is not sufficient for structural characterization. The authors should show that these two isolated products co-elute with authentic standards, and should also show that these isolated products have the same mass and retention time of the products of the in vitro reaction with PG (which is only demonstrated by a bar graph in Figure 4—figure supplement 1). High resolution mass spectra are very easy to obtain and are standard for molecular characterization and should also be included. These are fairly simple experiments and will provide much more convincing data that the structures are as indicated in Figure 4.

The authors state (subsection “*pltM* is necessary and sufficient for the *PltR*-mediated activation of *pltL* expression by PG”, first paragraph) "Moreover, *pltM* and *pltR* share similar transcriptional profiles (Clifford et al. 2015)." I was curious how the expression profile of *pltM* compared to rest of the *plt* and *phb* clusters. Can the authors make a brief comment on this, with the data that is currently available (i.e., the data in Clifford et al.)?

In the legend of Figure 6, the authors should indicate at what time point during the culture period these data points were taken.

---

## [Author Response]

*Essential revisions:*

*The reviewers were uniformly enthusiastic about publishing your article after some revision. Major issues that need to be addressed are summarized below, along with the individual review of reviewer #3.*

*1) Enhanced mechanistic analysis of the co-regulation including the difference/similarity of phlD and pltD, apparently redundant enzymes for PG production.*

We think there was some confusion surrounding this issue because *pltD* (gene locus ID: PFL_2790) is not mentioned in this manuscript and has no role in PG production. Previous studies showed that *phlD* (gene locus ID: PFL_5957), which encodes a type III polyketide synthase, is responsible for PG biosynthesis: heterologous expression of *phlD* led to an accumulation of PG in *Escherichia coli* cultures, and purified PhlD converted malonyl-coA into PG in vitro(Achkar et al. 2005). More importantly, *phlD* is the only gene for PG production of Pf-5, as a knockout of the *phlD* gene completely abolishes PG production (Kidarsa et al. 2011). To address this, we reiterated in the revised manuscript (page 4, line 66-67), that *phlD* is necessary for PG production.

*A mechanistic analysis would be facilitated by using "co-regulation", which is the much more informative description used in the impact statement, rather than "crosstalk", which following its more common definition of "components of different signal transduction pathways affecting one another", is inappropriate.*

Thank you for this suggestion. We changed all uses of “crosstalk” to “co-regulation”.

*2) The structural analysis of PG-Cl and PG-CL_2_ are not up to the usually accepted standards for establishing the structures of new molecules. It should be relatively easy to provide the additional data.*

We added the following additional evidence to support the structural analysis of PG-Cl and PG-Cl_2_.

First, by comparison to the synthesized authentic compounds PG-Cl and PG-Cl_2_, we detected PG- Cl and PG-Cl_2_ in the culture extracts of a Δ*gacA* mutant of Pf-5 containing a plasmid-borne *pltM*. These data were added as revised Figure 4 in the revised manuscript.

Second, by comparison to the synthesized PG-Cl and PG-Cl_2_, we detected these two compounds in the extracts of in vitro full reaction containing substrate PG, halogenase PltM and it co-factors, but not in the reaction without PltM. These data were added as Figure 5—figure supplement 1 in the revised manuscript.

*3) Suggest that the title be modified to: Metabolic co-regulation coordinates the biosynthesis of secondary metabolites in Pseudomonas protegens.*

Thank you. We changed the title to “Novel mechanism of metabolic co-regulation coordinates the biosynthesis of secondary metabolites in *Pseudomonas protegens*” as suggested by the Editor. We kept the “Novel mechanism” because of the novelty of the mechanism of co-regulation revealed in this study.

*Reviewer #3:*

*[…] I have the following comments below:*

*In Figure 5 only the mass of the products is shown, which is not sufficient for structural characterization. The authors should show that these two isolated products co-elute with authentic standards, and should also show that these isolated products have the same mass and retention time of the products of the in vitro reaction with PG (which is only demonstrated by a bar graph in Figure 4—figure supplement 1). High resolution mass spectra are very easy to obtain and are standard for molecular characterization and should also be included. These are fairly simple experiments and will provide much more convincing data that the structures are as indicated in Figure 4.*

We added the requested data to support the structural analysis of PG-Cl and PG-Cl_2_. We showed that PG- Cl and PG-Cl_2_ in the HPLC fraction 21 and 26 of culture extractions from the cultures of a Pf-5 derivative strain co-eluted with the synthesized PG-Cl and PG-Cl_2_ (Figure 4 in the revised manuscript).

Furthermore, by comparing the structures determined from extracts to those for the authentic compounds, we confirmed that PG-Cl and PG-Cl_2_ are present in the “full in vitro reaction” containing purified PltM and PG (Figure 5—figure supplement 1 in the revised manuscript). These data have been added to the revised manuscript.

*The authors state (subsection “pltM is necessary and sufficient for the PltR-mediated activation of pltL expression by PG”, first paragraph) "Moreover, pltM and pltR share similar transcriptional profiles (Clifford et al. 2015)." I was curious how the expression profile of pltM compared to rest of the plt and phb clusters. Can the authors make a brief comment on this, with the data that is currently available (i.e., the data in Clifford et al.)?*

The expression profile of *pltM, pltR* and the other genes in the pyoluteorin gene cluster was summarized from previous study (Clifford et al. 2015) and this information and supporting data were added in the revised manuscript (Figure 3—figure supplement 1).

*In the legend of Figure 6, the authors should indicate at what time point during the culture period these data points were taken.*

Thank you. GFP expression was tested at 20 hours after inoculation and the metabolites were extracted and tested at 24 hours after inoculation. This information has been added in the legend of Figure 6 of the revised manuscript.